# Mutant p53 induces Golgi tubulo-vesiculation driving a prometastatic secretome

Valeria Capaci [1], Lorenzo Bascetta [1,2,13], Marco Fantuz [1,2,13], Galina V. Beznoussenko[3], Roberta Sommaggio [4], Valeria Cancila[5], Andrea Bisso [1,12], Elena Campaner [1,6], Alexander A. Mironov[3], Jacek R. Wiśniewski[7], Luisa Ulloa Severino[6], Denis Scaini[2], Fleur Bossi[6], Jodi Lees[8], Noa Alon[8], Ledia Brunga[8], David Malkin [8,9], Silvano Piazza [1], Licio Collavin [1,6], Antonio Rosato[4,10], Silvio Bicciato[11], Claudio Tripodo[5], Fiamma Mantovani[1,6] & Giannino Del Sal [1,3,6✉]

*TP53* missense mutations leading to the expression of mutant p53 oncoproteins are frequent driver events during tumorigenesis. p53 mutants promote tumor growth, metastasis and chemoresistance by affecting fundamental cellular pathways and functions. Here, we demonstrate that p53 mutants modify structure and function of the Golgi apparatus, culminating in the increased release of a pro-malignant secretome by tumor cells and primary fibroblasts from patients with Li-Fraumeni cancer predisposition syndrome. Mechanistically, interacting with the hypoxia responsive factor HIF1α, mutant p53 induces the expression of miR-30d, which in turn causes tubulo-vesiculation of the Golgi apparatus, leading to enhanced vesicular trafficking and secretion. The mut-p53/HIF1α/miR-30d axis potentiates the release of soluble factors and the deposition and remodeling of the ECM, affecting mechano-signaling and stromal cells activation within the tumor microenvironment, thereby enhancing tumor growth and metastatic colonization.

[1] Laboratorio Nazionale CIB (LNCIB), 34149 Trieste, Italy. [2] International School for Advanced Studies (SISSA), 34146 Trieste, Italy. [3] Fondazione Istituto FIRC di Oncologia Molecolare (IFOM), 20139 Milan, Italy. [4] Veneto Institute of Oncology IOV-IRCCS, 35128 Padua, Italy. [5] Tumor Immunology Unit, Department of Health Science, Human Pathology Section, University of Palermo, School of Medicine, 90133 Palermo, Italy. [6] Dipartimento di Scienze della Vita, Università degli Studi di Trieste, 34127 Trieste, Italy. [7] Department of Proteomics and Signal Transduction, Max Planck Institute of Biochemistry, 85152 Martinsried, Germany. [8] Genetics and Genome Biology Program, The Hospital for Sick Children, Toronto, ON, Canada. [9] Department of Pediatrics, University of Toronto, Toronto, ON, Canada. [10] Department of Surgery, Oncology and Gastroenterology, University of Padova, 35128 Padova, Italy. [11] Center for Genome Research, University of Modena and Reggio Emilia, 41125 Modena, Italy. [12]Present address: Department of Experimental Oncology, IEO, European Institute of Oncology IRCCS, 20141 Milan, Italy. [13]These authors contributed equally: Lorenzo Bascetta, Marco Fantuz. ✉email: gdelsal@units.it

The crosstalk of cancer cells with the tumor microenvironment (TME) plays a crucial role in tumor growth and progression. Tumors reprogram their secretory phenotype to shape a permissive microenvironment, supporting local invasion and colonization of metastatic niches[1]. This involves the release of soluble factors stimulating angiogenesis and recruiting stromal and immune cell populations[2,3], as well as deposition and remodeling of the extra-cellular matrix (ECM) to sustain mechano-stimulation[4].

While many molecules and cell types involved in tumor–stroma communication have been characterized, factors responsible for reprogramming the secretory activity of tumor cells remain poorly understood. Several reports indicate that cancer cells adapt their secretory machinery, namely the endoplasmic reticulum (ER) and the Golgi apparatus (GA), to face increased protein translation and secretion[5,6]. Moreover, recent evidence suggests that alterations of the GA and optimization of vesicular trafficking endow cancer cells with aggressive and metastatic phenotypes[7,8]. How these alterations relate to oncogenic signaling is, however, largely unknown.

Missense mutations in the *TP53* gene, causing expression of mutant p53 proteins (mut-p53), are among most frequent genetic alterations in human cancers, and are associated with the Li-Fraumeni syndrome, a rare familial cancer predisposition[9,10]. Mut-p53 loses tumor suppressive functions and can acquire properties that enable it to rewire the cell's transcriptome and proteome, promoting tumor growth, chemoresistance, and metastasis[11–13]. Mut-p53 becomes frequently stabilized and activated by mechanical cues such as ECM stiffness[14], and impacts on the crosstalk between cancer cells and stroma by regulating the expression of cytokines and chemokines, thereby inducing tumor cell migration and invasion in a paracrine fashion[15]. However, the impact of mut-p53 on the secretory machinery and the effects of mut-p53-dependent secretome on TME at local and distal sites remain poorly defined. p53 missense mutants have been shown to regulate several miRNAs[16,17], some of which are secreted and concur to malignant evolution by long-range effects[18].

In this work we investigate how mut-p53 modifies cellular processes altering the communication of cancer cells with their microenvironment. As potential mediators of this activity we focused on mut-p53 regulated miRNAs, a class of molecules capable of modulating at multiple levels entire cellular processes. We discovered that mut-p53, through its target miR-30d, controls secretory trafficking in cancer cells by causing tubulo-vesiculation of the GA. This increases the release of a pro-malignant secretome, which impacts on TME, fostering tumor growth, and metastatic colonization.

## Results

**MicroRNA-30d is a novel target of mutant p53**. To identify mut-p53 target miRNAs, we silenced mut-p53[R280K] in MDA-MB-231 breast cancer cells by RNAi and monitored the levels of a panel of miRNAs, previously found overexpressed in solid tumor types at high frequency of missense *TP53* mutations[19,20] (Supplementary Fig. 1a). Among miRNAs whose expression was reduced upon mut-p53 knockdown we identified miR-30d, previously reported to exert oncogenic activities[21–25]. miR-30d expression was significantly reduced upon mut-p53 knockdown, while re-introduction of siRNA-insensitive p53[R280K] increased it (Fig. 1a). Similar effects were observed in cell lines harboring different mut-p53 variants, from breast (MD-MB-468/p53[R273H], SK-BR-3/p53[R175H], and SUM-159PT/p53[R158insS]) (Fig. 1b), prostate, colon, liver, and ovarian cancer (DU 145/p53[P223L,V274F], HT-29/p53[R273H], Mahlavu/p53[R249S], TOV-112D/p53[R175H], Supplementary Fig. 1b).

Silencing wild-type p53 had no effect on miR-30d levels in HBL100 and MCF-7 cancer cells, as well as in MCF10A normal-like breast epithelial cells (Fig. 1c, Supplementary Fig. 1c). Conversely, ectopic expression of mut-p53 variants R175H, R273H and R280K in MCF10A cells, stably silenced for wt-p53, increased miR-30d expression (Fig. 1c). Confirming dependence on mut-p53, miR-30d levels were reduced upon treating MDA-MB-231 cells with the mut-p53 inactivating agent APR-246/PRIMA-1MET, able to restore wt-p53 function[26] (Supplementary Fig. 1d). Moreover, uncoupling mechanosignaling by culturing cells on soft matrix, or by treatment with the myosin II inhibitor Blebbistatin or the HDAC6 inhibitor sulforaphane significantly reduced mut-p53 levels and miR-30d expression, similar to mut-p53 silencing (Supplementary Fig. 1d, e).

**Mut-p53 induces miR-30d expression through HIF1α**. We next investigated the mechanism of miR-30d regulation by mut-p53. qPCR analysis indicated that mut-p53 modulates the levels of primary (pri-miR-30d), precursor (pre-miR-30d), and mature miR-30d to similar extent (Fig. 1a, d), suggesting that it controls miR-30d transcription. The genomic regulatory regions of *MIR30D* are uncharacterized, however, the region surrounding the transcription start site[27,28] displayed active promoter chromatin marks[29] (Supplementary Fig. 1f), was identified as a target of mut-p53 by ChIP-sequencing[30], and is bound by Hypoxia-Induced Factors HIF1α and HIF2α under hypoxia[31]. This prompted us to test whether mut-p53 could regulate miR-30d transcription through HIFs. As shown in Supplementary Fig. 1g, ectopic expression of HIF1α increased miR-30d levels, while its depletion downregulated both precursor and mature miR-30d in MDA-MB-231 cells (Supplementary Fig. 1h). In addition, silencing either HIF1α or mut-p53 prevented the induction of miR-30d by hypoxia (2% pO$_2$) (Fig. 1e). Since mut-p53 transcriptional activity changes upon shifting cells from 2D to 3D culture[32], we confirmed that miR-30d induction by hypoxia was dependent on mut-p53 also when culturing MDA-MB-231 cells in 3D (Supplementary Fig. 1i).

Next, we verified whether mut-p53 and HIF1α interact: proximity ligation assays (PLA) revealed nuclear complexes between mut-p53 and HIF1α, whose formation was enhanced under low oxygen pressure as well as upon hypoxia-mimetic treatment with CoCl$_2$ (Fig. 1f). Similar results were obtained by co-immunoprecipitation of mut-p53 and HIF1α proteins from lysates of MDA-MB-231 cells grown under different oxygen pressure (Supplementary Fig. 1j), as well as from MDA-MB-468 cells, both in normoxia and upon hypoxia-mimetic treatment (Supplementary Fig. 1k).

Consistently, analysis of a breast cancer gene expression dataset[33] highlighted a correlation between *TP53* missense mutations and high HIF1α activity (Supplementary Fig. 1l). Moreover, high miR-30d expression was associated with *TP53* mutation and with high HIF1α levels in cancer gene expression datasets (Supplementary Fig. 1m, n).

Finally, to verify whether mut-p53 and HIF1α cooperatively activate the *MIR30D* promoter, we performed ChIP experiments in MDA-MB-231 cells. Under normoxia we observed binding of mut-p53 to *MIR30D* promoter, that was further enhanced under hypoxia in a HIF1α-dependent manner (Fig. 1g). On the other hand, mut-p53 was required for efficient recruitment of HIF1α to *MIR30D* promoter under normoxia and hypoxia (Fig. 1h), and for histone H3 Lysine 9 acetylation (H3K9ac) (Fig. 1i), which occurs during hypoxia-induced transcriptional activation[34] at this genomic region.

These results suggest that in cancer cells, mut-p53 and HIF1α form an active transcriptional complex on *MIR30D* promoter, leading to miR-30d expression already in normoxic conditions, further increased by hypoxia.

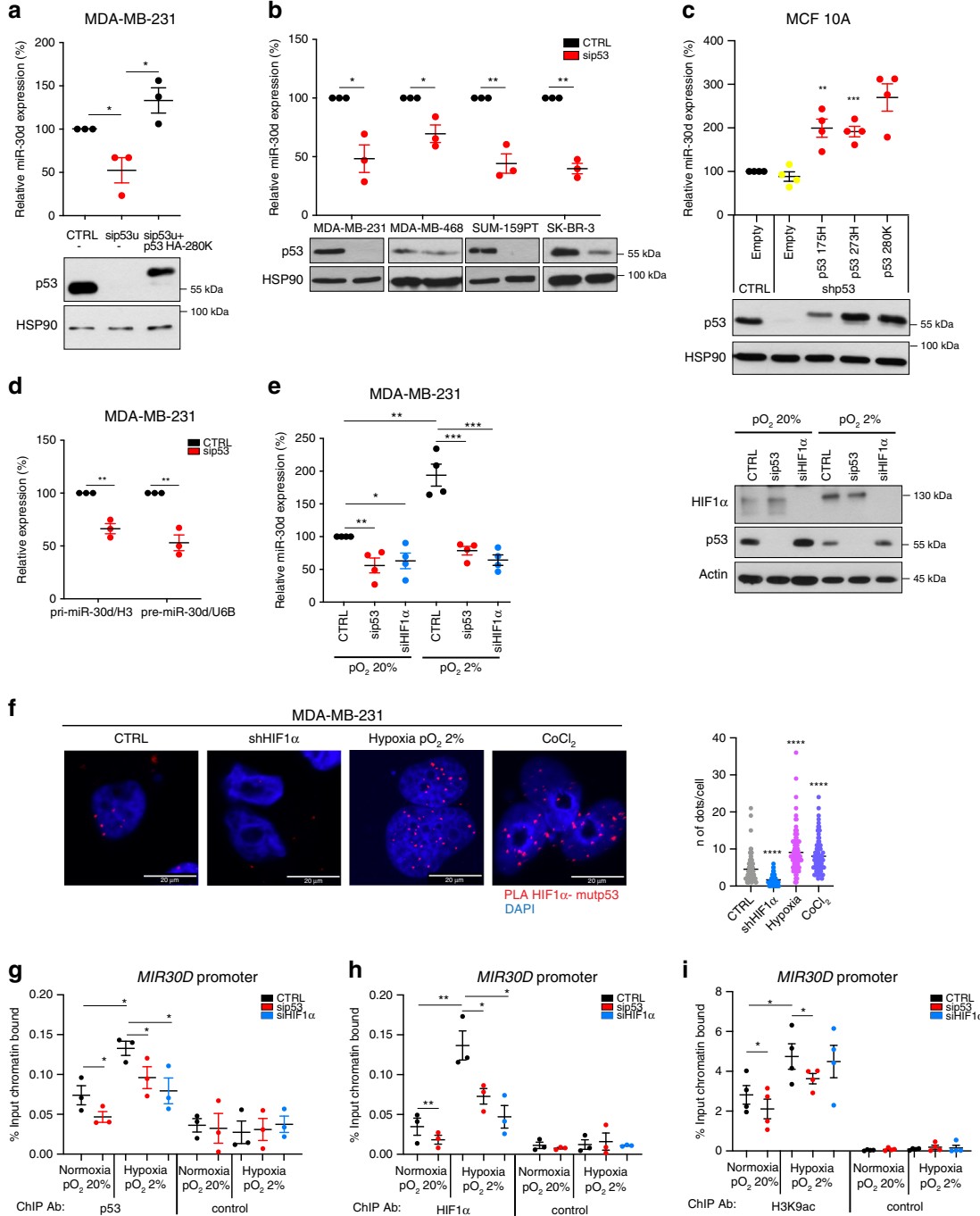

**Fig. 1 Mutant p53 induces miR-30d expression through HIF1α. a** miR-30d expression was evaluated by RT-qPCR, normalized to U6B RNA expression levels, in MDA-MB-231 cells upon silencing endogenous mut-p53 with a siRNA targeting the 3′UTR (sip53u); expression of mut-p53 R280K was rescued by transfecting a siRNA-resistant mut-p53 HA-R280K construct. Bottom: western blot analysis of p53 expression using HSP90 as loading control. **b** Expression levels of miR-30d were analyzed as in (**a**) upon silencing of mut-p53 in the indicated human breast cancer cell lines. **c** Endogenous wt-p53 was stably silenced in MCF10A mammary epithelial cells (shp53), and shRNA-resistant forms of the indicated p53 mutants were expressed by viral transduction where indicated. miR-30d expression was then evaluated as in (**a**). **d** mut-p53 was silenced in MDA-MB-231 cells as in (**a**); expression of pri-miR-30d and pre-miR-30d was then evaluated by RT-qPCR, normalized, respectively, to the expression of H3 and U6B RNA. **e** Expression of miR-30d was evaluated by RT-qPCR as in (**a**), upon silencing of either mut-p53 or HIF1α in MDA-MB-231 cells cultured either in normoxic (20% pO₂) or hypoxic conditions (2% pO₂) for 16 h. Right: western blot analysis of HIF1α and mut-p53 levels, using actin as a loading control. **f** Proximity ligation assay (PLA) with primary antibodies against p53 and HIF1α were performed in MDA-MB-231 cells cultured either under normoxic (20% pO₂) or hypoxic conditions (2% pO₂) for 16 h, or exposed for 16 h to 150 μM CoCl₂ as hypoxia-mimetic treatment. **g–i** Lysates of MDA-MB-231 cells cultured under the indicated conditions for 16 h upon silencing of either mut-p53 or HIF1α were subjected to chromatin immunoprecipitation (ChIP) analysis with anti-p53 FL-393 (**g**), HIF1α (**h**), and Acetyl-Histone H3 (Lys9) antibodies (**i**), or Protein A/G PLUS-Agarose as negative control. Binding to *MIR30D* promoter region was calculated as fraction of input chromatin bound. Binding to non-specific chromatin is shown in Supplementary Fig. 1o-p-q. Graphs represent the individual data points, the mean +/− SEM of three independent experiments. Blots are representative of n = 3 biological replicates. P value (*p < 0.05, **p < 0.01, ***p < 0.001) was calculated by two-tailed unpaired Student's t-test.

**miR-30d regulates genes involved in the secretory pathway**. At first, we analyzed the impact of the mut-p53/miR-30d axis on cell transformation in vitro. Consistently with previous reports[32,35], MCF10A cells exhibited normal acinar morphogenesis in 3D cultures, while ectopic expression of mut-p53 variants R175H or R280K, but not wt-p53 silencing per se[36], inhibited luminal clearance, reminiscent of the filled lumen phenotype of ductal carcinoma in situ (Supplementary Fig. 2a). In this context, inhibition of miR-30d by a decoy vector (dy-30d)[37] abolished luminal filling (Supplementary Fig. 2a), suggesting that miR-30d mediates oncogenic phenotypes downstream to mut-p53.

To identify cellular processes regulated by miR-30d, we performed gene expression analysis in MDA-MB-231 cells stably transduced with dy-30d (hereafter referred to as MDA-MB-231/dy-30d). Functional annotation analysis revealed that differentially expressed genes mainly belong to functional categories of protein transport, vesicular trafficking and Golgi organization (Fig. 2a; Supplementary Data File 1). Gene Set Enrichment Analysis (GSEA) indicated that genes regulated by miR-30d are involved in protein secretion (Fig. 2b). We validated a panel of these targets by qRT-PCR upon miR-30d overexpression in MDA-MB-231 cells. As shown in Supplementary Fig. 3b, miR-30d enhanced the expression of key components of ER-related transport (e.g., SEC11C, SSR1) and of ER-Golgi vesicular trafficking machinery (e.g., SEC23A, SEC24A, SEC24B, COPB1, ARF4, ARFGEF1, GOSR2), while reducing the expression of negative regulators of ER-Golgi trafficking (e.g., ARFGAP2) and kinesin-mediated retrograde transport (e.g., KIF20A). In sum, gene expression data suggest that miR-30d could modulate secretory trafficking.

We then asked if miR-30d affects protein secretion. MDA-MB-231/dy-30d and control cells were metabolically labeled with $^{S35}$Met/Cys aminoacids, and proteins secreted in the conditioned medium (CM) were analyzed by SDS-PAGE and autoradiography. As shown in Fig. 2c, miR-30d depletion significantly decreased total protein secretion. This was not due to reduced protein synthesis, as levels of intracellular proteins were not appreciably affected (Supplementary Fig. 3c). mut-p53 knockdown similarly reduced secretion, that was rescued by concomitant miR-30d overexpression (Fig. 2d, Supplementary Fig. 3d).

These results were recapitulated in mut-p53$^{R273H}$ expressing MDA-MB-468 cells (Supplementary Fig. 3e), and in isogenic MCF10A-derived cells expressing either mut-p53$^{R175H}$ or mut-p53$^{R280K}$ (Supplementary Fig. 3f), indicating that different p53 mutants share the ability to control secretory trafficking, while knockdown of wt-p53 did not significantly affect protein secretion (Supplementary Fig. 3f). Finally, we verified that HIF1α knockdown dampened protein secretion, while HIF1α stabilization by CoCl$_2$ promoted it (Supplementary Fig. 3g), in line with the observation that HIF1α upregulates miR-30d.

We then sought to confirm these results using a reporter for canonical protein secretion[38] (ssGFP), that localizes along the whole secretory pathway (Supplementary Fig. 4a). In line with above results, mut-p53 knockdown reduced the amount of secreted ssGFP in MDA-MB-231 cells, while miR-30d overexpression reversed the effect both in 2D and 3D cultures (Fig. 2e and Supplementary Fig. 4b). Similar results were obtained in DU 145 (mut-p53$^{P223L, V274F}$), HT-29 (mut-p53$^{R273H}$), and Mahlavu (mut-p53$^{R249S}$) cells (Supplementary Fig. 4c–e). Similarly, ssGFP secretion was dampened upon mut-p53 inactivation with APR-246/PRIMA-1MET, or by reducing mut-p53 stability interfering with mechanosignaling (Supplementary Fig. 4f). As shown in Fig. 2f, inhibition of miR-30d with dy-30d strongly reduced the ability of mut-p53$^{R280K}$ to enhance ssGFP secretion in MCF10A cells. Finally, silencing mut-p53 in MDA-MB-231 cells transfected

with a miR-30d-inhibitor hairpin (IH-30d) did not lead to significant further reduction of ssGFP secretion (Supplementary Fig. 4g), confirming that in these conditions miR-30d is a major mediator of the effect of mut-p53 on secretion.

Consistently with the observed induction of miR-30d by HIF1α, knockdown of HIF1α dampened ssGFP secretion, whereas hypoxia promoted it in a miR-30d-dependent manner (Fig. 2g).

Altogether, these results suggest that the mut-p53/HIF1α/miR-30d axis stimulates protein secretion in cancer cells.

**The mut-p53/miR-30d axis impacts cell secretory machinery**. To characterize the proteins whose secretion is stimulated by mut-p53/miR-30d, we performed mass spectrometry analysis by LC–MS/MS technology on CMs collected from control and mut-p53-KD MDA-MB-231 cells, and from same cells overexpressing miR-30d (details in Supplementary Fig. 4h and Methods).

As shown in Fig. 2h, mut-p53 knockdown altered significantly the protein secretome of MDA-MB-231 cells, with both up- and downregulation of a large number of hits; notably, overexpressing miR-30d largely reverted the effects of mut-p53 knockdown.

We then compared mut-p53 secretome data with mut-p53-dependent transcriptome and proteome data previously obtained in the same cell line[30]. This revealed that only about 30% (247/815) of differentially secreted proteins were regulated also at the transcript or protein level upon mut-p53 knockdown (Fig. 2i), with a striking discrepancy between miR-30d-dependent secretome and transcriptome (only 11% hits in common, 107/988) (Fig. 2j). This indicates that the effect of mut-p53/miR-30d on protein secretion depends only in part on altered expression of secreted proteins, implying that it could impinge on the secretion process.

Hence, we analyzed the secretory and trafficking pathway in MCF10A cells, upon silencing wt-p53 and overexpressing either mut-p53$^{R280K}$ or miR-30d, by monitoring marker expression and organelle morphology relative to ER, GA, COPI-II transport vesicles, and microtubules. Levels of PDIA5, Sec24A, and GM130 proteins (markers for ER, COPI, and cis-Golgi compartments, respectively) were increased upon mut-p53$^{R280K}$ and miR-30d overexpression, while silencing wt-p53 did not affect them (Supplementary Fig. 5a). Immunofluorescence and confocal microscopy revealed that, while wt-p53 depletion did not alter the structure of any component of the secretory pathway, expression of either mut-p53$^{R280K}$ or miR-30d (Fig. 3a) caused strong alterations of GA morphology, with only mild effects on ER morphology (PDIA5), COP vesicles (Sec24A and β-COP), and microtubules (α-tubulin). In more than 60% of miR-30d overexpressing cells, the GA perinuclear ribbon-like structure was replaced by multiple mini-stacks dispersed within the cytoplasm (Fig. 3a and Supplementary Fig. 5b), a morphology hereafter defined as vesiculation. This phenotype was confirmed by staining for HPA (a lectin that binds glycans processed in cis-Golgi cisternae), Giantin (an intercisternal cross-bridges marker), and TGN46 (a trans-Golgi marker) (Supplementary Fig. 5c). Consistently, 3D image reconstitution using confocal Z-stacks revealed an increase from about 10 to 60 cis-Golgi elements per cell upon miR-30d overexpression (Supplementary Fig. 5d and Movie 1).

Of note, GA alteration induced by mut-p53 in MCF10A cells was reverted upon miR-30d inhibition by dy-30d (Fig. 3b). Similarly, ectopic expression of mut-p53 variants R175H, R273H, and R280K in p53-null H1299 lung cancer cells induced GA vesiculation that was abrogated by inhibiting miR-30d (Supplementary Fig. 5e). Moreover, MDA-MB-231 and Mahlavu cells displayed a vesiculated GA morphology that was normalized upon knockdown of mut-p53 or treatment with PRIMA-1MET,

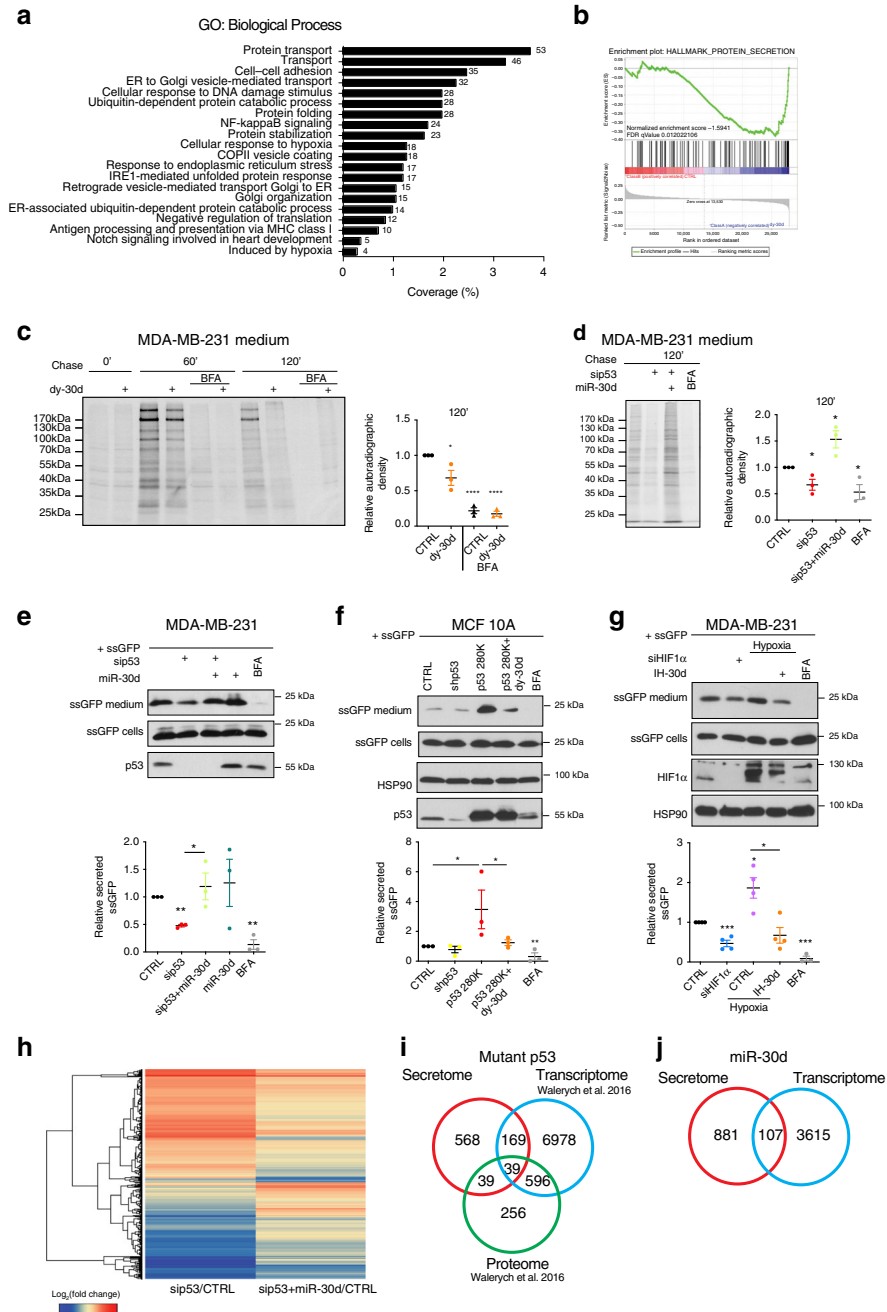

**Fig. 2 mut-p53 and miR-30d regulate protein secretion. a** Gene Ontology (GO) enrichment analysis of genes differentially expressed in MDA-MB-231 cells transduced with miR-30d decoy compared with control, using DAVID database. All terms were significant ($p < 0.05$) following Benjamini–Hochberg correction. **b** Gene set enrichment analysis (GSEA) of hallmarks PROTEIN_SECRETION in MDA-MB-231-dy-30d transcriptome compared with control ($n = 3$). **c** Analysis of protein secretion in MDA-MB-231-dy-30d cells by [$^{35}$S]-methionine/cysteine labeling, SDS-PAGE, and autoradiography (top). The inhibitor of ER-GA protein transport Brefeldin A (BFA, 2.5 µM) was used as control. Intracellular proteins are shown in Supplementary Fig. 3c. Right: ratio of secreted vs intracellular labeled proteins quantified by densitometry. **d** Protein secretion analysis in MDA-MB-231 cells transfected with mut-p53 siRNA, miR-30d mimic or combination, performed as in (**c**) (intracellular proteins are in Supplementary Fig. 3d). **e** MDA-MB-231 cells were transduced with construct encoding secretion signal-fused GFP (ssGFP). Forty-eight hours upon silencing mut-p53, overexpressing miR-30d mimic or combination, fresh medium was added and collected after 2 h. Bottom: ratio of secreted versus intracellular ssGFP quantified by densitometry. **f** Intracellular and medium (CM) ssGFP levels were analyzed as in (**e**) in MCF10A cells silenced for wt-p53 (shp53) and overexpressing mut-p53$^{R280K}$, and transduced with dy-30d or control. **g** Intracellular and CM ssGFP levels were analyzed as in (**e**) in MDA-MB-231 cells upon silencing HIF1α or culturing in hypoxia (2% pO$_2$) for 16 h, and treated or not with miR-30d inhibitor. **h** MDA-MB-231 cells were transfected as in (**d**), and CM was analyzed by LC–MS/MS. Heatmap shows proteins whose secretion was significantly altered. **i** Venn diagram showing overlap between proteins differentially secreted (**h**: secretome) upon mut-p53 silencing in MDA-MB-231 cells and genes differentially expressed at protein (proteome) or RNA level (transcriptome) as reported in ref. [30]. **j** Venn diagram showing overlap between differentially secreted proteins (secretome) and differentially expressed transcripts (transcriptome) upon inhibiting miR-30d in MDA-MB-231 cells by decoy construct (dy-30d). Graphs represent the individual data points, the mean $+/-$ SEM of three independent experiments. Blots are representative of $n = 3$ biological replicates. $P$ value (*$p < 0.05$, **$p < 0.01$, ***$p < 0.001$) was calculated by two-tailed unpaired Student's $t$-test. Source data are provided as Source data file.

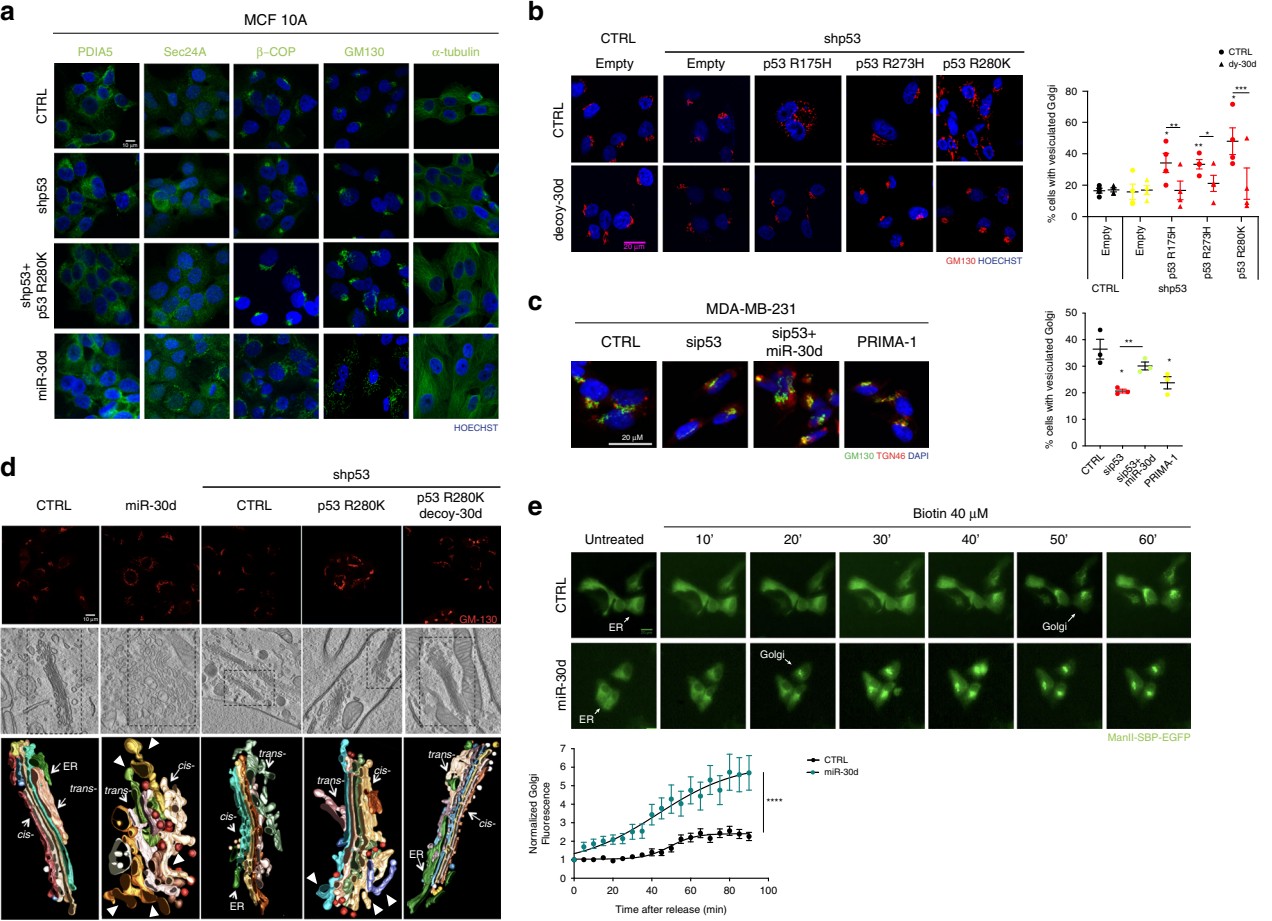

**Fig. 3 The mut-p53/miR-30d axis modifies the cell secretory machinery. a** Immunofluorescence analysis of PDIA5 (ER), SEC24 (COPII), βCOP (COPI), GM130 (*cis*-Golgi), and α-tubulin (microtubules) in MCF10A cells, upon stable silencing of endogenous wild-type p53 (shp53) either alone or combined with overexpression of mut-p53 R280K or miR-30d mimic. Nuclei were stained with Hoechst. Scale bar, 20 μm. **b** Immunofluorescence analysis of Golgi apparatus stained with an antibody specific for GM130 in MCF10A cells stably silenced for endogenous p53 (shp53) and overexpressing mut-p53 R175H, R273H, or R280K forms, and transduced either with control or miR-30d decoy construct. Scale bar, 20 μm. Right: graph showing the percentage of cells with vesiculated Golgi upon different treatments. **c** Immunofluorescence analysis of Golgi apparatus stained with an antibody specific for GM130 in MDA-MB-231 cells upon silencing of mut-p53, overexpression of a miR-30d mimic, their combination or treatment for 24 h with 10 μM PRIMA-1 as indicated. Scale bar, 20 μm. Right: graph showing the percentage of cells with vesiculated Golgi upon different conditions. **d** Top: confocal superresolution microscopy analysis of Golgi structure in MCF10A cells subjected to the indicated treatments, and stained for GM130 antigen. Middle: electron tomography of the above samples. Bottom: three-dimensional model of the Golgi stacks highlighted in the middle panels, showing the Golgi cisternae. Arrowheads indicate vesicular-tubular clusters in cells overexpressing miR-30d or mut-p53 R280K. **e** Images show the cellular localization of the RUSH reporter MannII-SBP-EGFP at the indicated time points after addition of biotin in MCF10A cells transfected with either CTRL or miR-30d mimic. Scale bar, 20 μm. Bottom: plots show total fluorescence intensity in the Golgi region at each time point, corrected for background and normalized to the fluorescence before release. Curves depict the measurement of 20 cells for each condition. Time per condition *p* value <0.0001 was calculated by two-way ANOVA test. In (**b**–**d**) graphs represent the individual data points and the mean +/− SEM of three independent experiments. *P* value (**p* < 0.05, ***p* < 0.01, ****p* < 0.001) was calculated by two-tailed unpaired Student's *t*-test). Micrographs are representative of *n* = 3 biological replicates. Source data are provided as Source data file.

and reverted by introduction of miR-30d (Fig. 3c and Supplementary Fig. 5f).

These results confirm that mut-p53 modifies GA morphology and miR-30d is a major mediator of this activity. Consistently with the observed regulation of miR-30d by HIF1α, activation of HIF1α by hypoxia or treatment with CoCl₂ caused a similar modification of GA structure that was reduced by miR-30d inhibition (Supplementary Fig. 5g).

To characterize GA alterations induced by the mut-p53/miR-30d axis, we performed ultrastructural analysis by correlative light electron microscopy (CLEM) in MCF10A cells. While control cells exhibited the canonical GA ultrastructure, miR-30d over-expressing cells displayed tubulo-vesiculated GA, characterized by narrow cisternae with large pores, intercisternal connections,

membrane invaginations, and swelling of cisternae (Fig. 3d). An increased number of COPI vesicles was also detected. CLEM analysis indicated that depletion of wt-p53 did not alter GA structure, while mut-p53^R280K overexpression caused appearance of partially tubulated cisternae (Fig. 3d). Of note, miR-30d inhibition restored the normal structure of GA stacks in mut-p53 overexpressing cells (Fig. 3d), confirming epistatic relationship. Single cisternae reconstitution is shown in Supplementary Fig. 5h-l.

Alterations of GA structure similar to those induced by mut-p53/miR-30d have been previously reported to accelerate diffusion of soluble proteins from *cis*- to *trans*-Golgi, leaving unaffected or reducing the trafficking of non-diffusible cargoes including membrane proteins[39,40]. Moreover, the observed increase of COPI

vesicles suggests that enhanced retrieval of ER and Golgi resident proteins may occur to maintain the correct intracisternae distribution of Golgi enzymes. To analyze alterations of protein transport kinetics induced by mut-p53/miR-30d we employed the RUSH reporter system, which allows to synchronize ER exit of ectopically expressed proteins[41]. We transfected as reporter the α-mannosidase II (MannII) fused to streptavidin binding domain together with a Streptavidin-KDEL construct as a hook for ER retention. MannII allows to follow the traffic from ER to *medial-trans*-Golgi[42,43]. As shown in Fig. 3e, miR-30d overexpression associated with faster Golgi localization of MannII (20 min upon synchronization vs 50 min in control cells).

These results demonstrate that mut-p53, via miR-30d, modifies the secretory machinery by inducing tubulo-vesiculation of the Golgi apparatus and increases trafficking rate.

**miR-30d regulates targets involved in the secretory pathway.** We sought to identify candidate miR-30d targets that could mediate the observed phenotypes. Having shown that miR-30d expression is induced by mut-p53 and HIF1α, we used available transcriptomic data to compare three gene lists: (i) genes downregulated by mut-p53 (sip53 UP, i.e., genes upregulated by mut-p53 KD in MDA-MB-231 cells[30], (ii) genes downregulated by HIF1α[33] (siHIF1α UP), and (iii) genes downregulated by miR-30d (dy-30d UP, i.e., genes upregulated by dy-30d in MDA-MB-231 cells). Intersection of these sets returned 118 genes commonly regulated by mut-p53/HIF1α/miR-30d; ten of these mRNAs are predicted miR-30d targets in silico (www.targetscan.org) (Fig. 4a). Validation by RT-qPCR confirmed that mut-p53 represses AP2A1, DGKZ, IQCG, PPP3CB, and VPS26B through miR-30d (Fig. 4b and Supplementary Fig. 6a). Interestingly, these genes have been associated to vesicular trafficking and recycling processes. We analyzed the ability of miR-30d to directly downregulate these mRNAs by 3′UTR-luciferase reporter assays (Fig. 4c–d, Supplementary Fig. 5b). We observed reduction of reporter activity only for DGKZ and VPS26B constructs, that was prevented by introducing targeted mutations at predicted miR-30d binding sites, suggesting that DGKZ and VPS26B are direct targets of miR-30d (Fig. 4d). Consistently, endogenous DGKZ and VPS26B proteins increased upon mut-p53 depletion in MDA-MB-231 cells, while miR-30d overexpression reverted this effect (Supplementary Fig. 6c). Similarly, hypoxic conditions reduced DGKZ and VPS26B transcripts, while HIF1α knockdown increased them (Supplementary Fig. 6d).

DGKZ is an enzyme belonging to the diacylglycerol kinase family[44], while VPS26B is a component of the retromer core complex, which mediates the recycling of proteins during endosomal sorting[45,46]. VPS26B and its paralogue VPS26A represent mutually exclusive subunits that define distinct retromer complexes[47]. Of note, mut-p53 and miR-30d regulate specifically VPS26B expression, while not affecting VPS26A (Supplementary Fig. 6e).

We next investigated the role of DGKZ and VPS26B in GA morphology. As shown in Fig. 4e, downregulation of either DGKZ or VPS26B by RNAi induced GA vesiculation in MCF10A cells, mimicking miR-30d overexpression. We then analyzed protein secretion by metabolic labeling: depletion of DGKZ or VPS26B in MCF10A cells resulted in significant increase of total protein secretion, similarly to miR-30d overexpression (Fig. 4f, Supplementary Fig. 6f).

By catalyzing the conversion of membrane lipid diacylglycerol (DAG) in phosphatidic acid, DGKZ decreases the levels of DAG at membranes[44]. The local concentration of DAG regulates Golgi tubulo-vesiculation[48] and promotes trafficking from *trans*-Golgi to plasma membrane via activation of Golgi resident PKD kinase[49,50]. We thus hypothesized that downregulation of DGKZ

by miR-30d may lead to DAG accumulation at Golgi membranes inducing local PKD activation. We tested this possibility using a fluorescent probe consisting of the DAG-binding domain of PKCγ fused to GFP[51]. In MDA-MB-231 cells, mut-p53 knockdown reduced DAG levels at *cis-* and *trans*-Golgi, while concomitant miR-30d overexpression rescued DAG accumulation (Supplementary Fig. 6g). We then monitored PKD1 activation upon mut-p53 knockdown, observing a reduction of the active phosphorylated form of PKD1 both in total cell lysates and specifically at Golgi membranes as indicated by immunofluorescence and confocal imaging, while either miR-30d overexpression or DGKZ knockdown reversed this effect (Fig. 4g). It is thus conceivable that DAG accumulation and consequent activation of PKD contribute to the ability of mut-p53/miR-30d to enhance secretion.

miR-30d has been previously reported to reduce expression of the Golgi GalNAc transferase GALNT7, affecting O-glycosylation and promoting invasion and metastasis of melanoma cells[21]. We confirmed that miR-30d reduced GALNT7 expression also in MDA-MB-231 cells (Supplementary Fig. 6i). However, GALNT7 depletion in MCF10A cells did neither cause GA vesiculation (Supplementary Fig. 6j), nor enhanced ssGFP secretion (Supplementary Fig. 6k), suggesting that its downregulation does not contribute to GA alteration by miR-30d.

Altogether, these findings identify DGKZ and VPS26B as direct targets of mut-p53/miR-30d axis, whose inhibition contributes to GA vesiculation and increased secretion. Moreover, these data suggest that DGKZ inhibition by miR-30d influences GA morphology and function by altering DAG-dependent signaling.

Interestingly, also indirect effects of miR-30d, particularly microtubule stabilization (as judged by staining for acetylated tubulin shown in Supplementary Fig. 6l), may promote secretion. In fact, the microtubule stabilizing agent Taxol, which increases Golgi vesiculation and protein secretion (Supplementary Fig. 6l-m), partially rescues ssGFP secretion upon miR-30d inhibition in MDA-MB-231 cells (Supplementary Fig. 6m).

**Mut-p53/miR-30d secretome impacts on tumor microenvironment.** We then investigated the functional relevance of altered secretion by mut-p53/miR-30d. Bioinformatic analysis of mut-p53/miR-30d secretome (Fig. 5a) revealed enrichment of ECM composition and remodeling, cell–ECM and cell–cell adhesion, angiogenesis and migration functional categories (Fig. 5a, Supplementary Data File 2).

This prompted us to explore whether secretome reprogramming could affect the crosstalk of cancer cells with their microenvironment. In particular, altered ECM deposition and remodeling increases stromal stiffness and affects mechanosignaling, altering the activity of CAFs and other cells populating the TME[4]. Immunoblot analysis of MDA-MB-231 CM highlighted that mut-p53 knockdown reduced secreted ECM components including Fibronectin, Laminin V and Laminin B1, while miR-30d overexpression increased them (Fig. 5b, Supplementary Fig. 7a). Accordingly, using atomic force microscopy we found that mut-p53 knockdown in MDA-MB-231 cells decreased ECM rigidity (0.15 kPa vs 0.25 kPa), while miR-30d overexpression induced a stiffer ECM (0.30 kPA) (Fig. 5c and Supplementary Fig. 7b).

Next, we evaluated paracrine effects of mut-p53/miR-30d secretome on tumor and stromal cells, including fibroblasts and endothelial cells. To evaluate the impact on cancer cell migration, H1299 lung cancer cells were treated with CMs collected from either H1299 or MDA-MB-231 cells. As shown in Fig. 5d, CM from MDA-MB-231 cells increased migration of H1299 cells more than twofold, an effect significantly reduced upon depleting

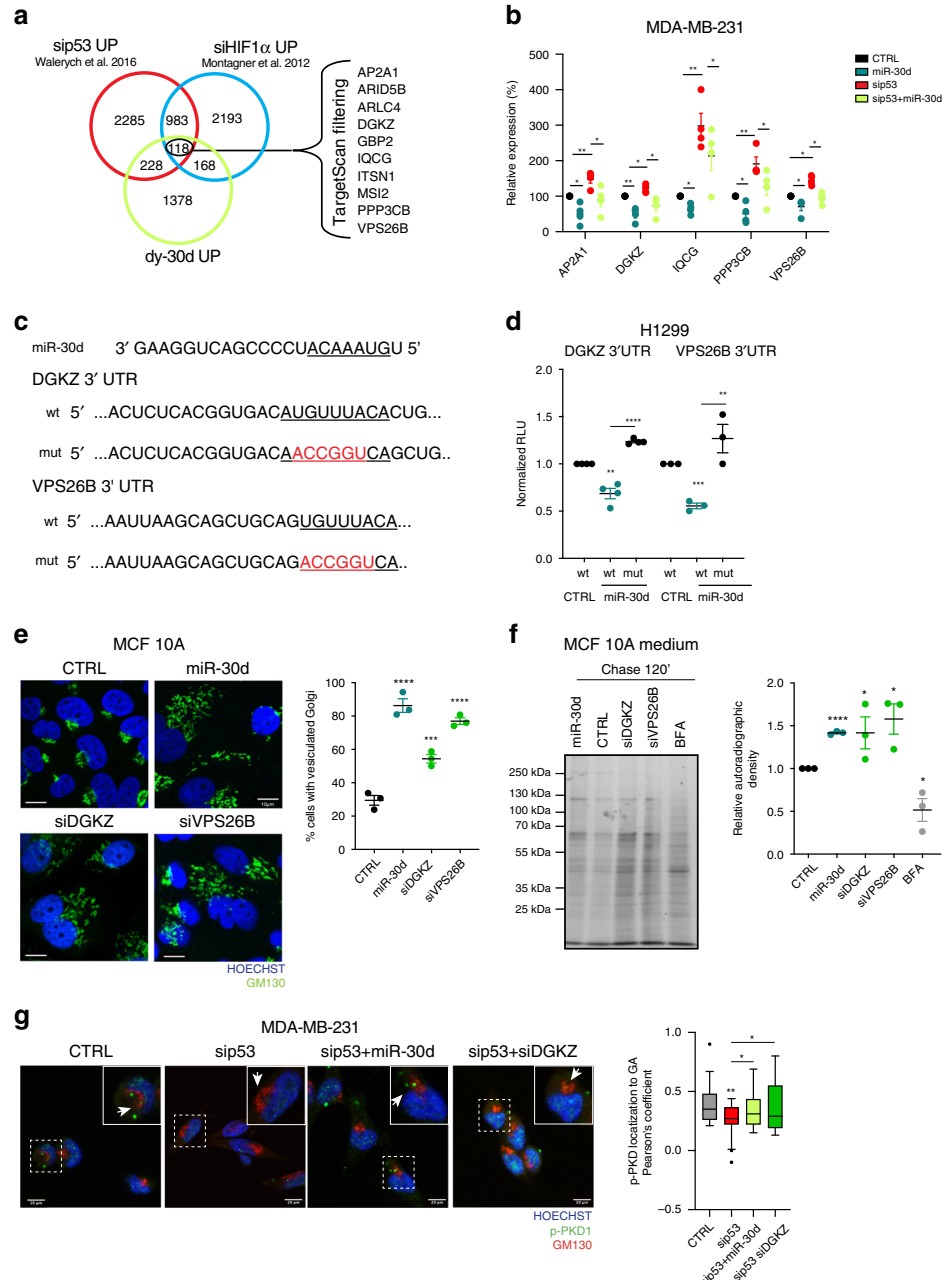

**Fig. 4 miR-30d alters the secretory pathway via DGKZ and VPS26B. a** Venn diagram illustrating the overlap between transcripts upregulated by miR-30d inhibition (decoy-miR-30d UP), mut-p53 depletion (sip53 UP[30]), and HIF1α depletion (siHIF1a UP[33]) in the MDA-MB-231 cell line. Common genes that are predicted miR-30d targets by TargetScan algorithm are listed. **b** Expression of indicated mRNAs in MDA-MB-231 cells transfected with either miR-30d mimic, mut-p53 siRNA, or their combination was analyzed by RT-qPCR, normalized to H3 expression. **c** Top: miR-30d sequence. Bottom: sequences of the 3' UTR regions of *DGKZ* and *VPS26B*. The miR-30d seed sequences are underlined, mutagenized sequences in the constructs are indicated in red. **d** Luciferase assays were performed in H1299 cells with reporter plasmids as described in (**c**); reporter plasmids were co-transfected with control (CTRL) or miR-30d expression constructs. Firefly luciferase expressed by the same plasmid was used to normalize for transfection efficiency. **e** Immunofluorescence analysis of Golgi morphology by staining for GM130 in MCF10A cells upon overexpressing miR-30d mimic or silencing DGKZ or VPS26B. Nuclei were stained with Hoechst. Scale bar, 20 µm. The graph shows the percentage of cells with vesiculated Golgi structure upon different treatments. **f** Analysis of total protein secretion in MCF10A cells after overexpression of miR-30d mimic, or silencing of DGKZ or VPS26B, and metabolically labeled with [35S]-methionine/cysteine aminoacids. Labeled proteins were resolved by SDS-PAGE and detected by autoradiography (representative of n = 3 biological replicates). Treatment with 2.5 µM BFA was used as a control. Right: graph showing the ratio of secreted versus intracellular labeled proteins quantified by densitometry (corresponding intracellular proteins are shown in Supplementary Fig. 4f). **g** Co-localization of phosphorylated PKD (pPKD) with the Golgi marker GM130 in MDA-MB-231 cells upon mut-p53 knockdown (sip53), and either introduction of miR-30d mimic or DGKZ silencing (siDGKZ). Scale bar, 20 µm. The bloxplot shows the Pearson's correlation coefficient for pPKD Golgi localization (n = 25 cells for each condition), median and whiskers calculated with Tukey method. Graphs represent the individual data points, mean +/− SEM of three independent experiments. P value (*p < 0.05, **p < 0.01, ***p < 0.001, ****p < 0.0001) was calculated by two-tailed unpaired Student's t-test. Source data are provided as Source data file.

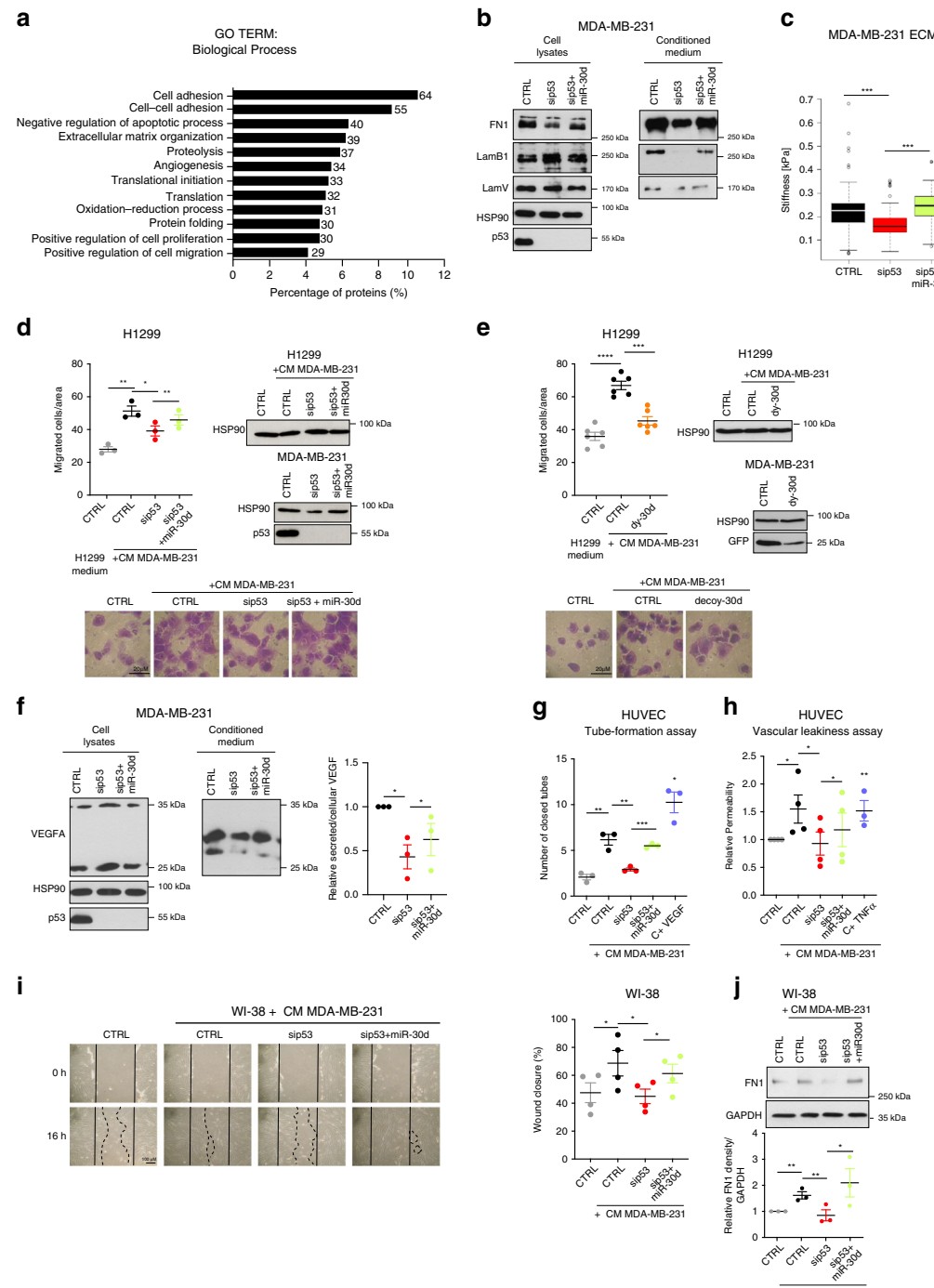

**Fig. 5 The mut-p53/miR-30d secretome induces a supportive TME. a** GO enrichment analysis of differentially secreted proteins (Fig. 2g data), using DAVID database. All terms were significant ($p < 0.05$ Benjamini–Hochberg correction). **b** Analysis of intracellular (cell lysates) and secreted (CM) proteins in MDA-MB-231 cells transfected with mut-p53 siRNA (sip53) alone or with miR-30d mimic. **c** AFM measurement of the stiffness of ECM deposited by MDA-MB-231 cells transfected with mut-p53 siRNA alone or with miR-30d mimic. Boxplot represents median and whiskers (Tukey method). **d** H1299 cells were treated with H1299-conditioned DMEM or conditioned DMEM (CM) from MDA-MB-231 cells transfected as in (**b**); cell migration was analyzed by transwelling (Boyden chamber) assays (16 h). Right: cell amounts and mut-p53 levels were compared by western blot. Bottom: Representative light microscope images of migrated cells. **e** H1299 cell migration analysis performed as in (**d**), using CM from MDA-MB-231 cells transduced with control (CTRL) or dy-30d. Control of decoy vector was GFP. **f** Levels of intracellular and secreted VEGFA analyzed as in (**b**). Right: relative VEGFA secretion quantified by densitometry. **g** Analysis of endothelial network formation by HUVEC cells on Matrigel upon addition of CM from MDA-MB-231 cells transfected as in (**b**). The graph shows numbers of endothelial closed loops ($n = 4$). VEGF (20 ng/ml for 24 h) was used as control. **h** Dextran permeability assay of HUVEC monolayers was performed upon addition in the upper chamber of MDA-MB-231 CM as in (**b**). FITC-Dextran fluorescence in the lower chamber was measured after 30′. TNFα (100 ng/ml for 30 min) was used as control. **i** Representative light microscopy images of wound healing assays of WI-38 fibroblasts treated with their medium (CTRL) or with MDA-MB-231 CM as in (**d**). The graph shows percentage of wound closure. **j** Immunoblot analysis of fibronectin1 in WI-38 cells treated as in (**i**). The blot is representative of $n = 3$ biological replicates. Bottom: densitometry quantification of FN1 expression relative to GAPDH. Graphs represent individual data points and mean $+/-$ SEM; $p$ value (*$p < 0.05$, **$p < 0.01$. ***$p < 0.001$, ****$p < 0.0001$) calculated by two-tailed unpaired Student's $t$-test. Blots are representative of $n = 3$ biological replicates. Source data are provided as Source data file.

mut-p53 in donor cells. Overexpression of miR-30d was sufficient to restore a pro-migratory secretion in mut-p53-KD MDA-MB-231 cells (Fig. 5d). Similarly, inhibition of miR-30d abolished the pro-migratory effect of MDA-MB-231 CM on H1299 cells (Fig. 5e). Of note, this property relied on the protein component of the secretome, since it was abolished by heat denaturation (Supplementary Fig. 7c).

Next, we analyzed the impact of mut-p53/miR-30d secretome on endothelial cells and fibroblasts. Mut-p53 directly promotes VEGFA transcription[52] (Supplementary Fig. 7d), and consistently with secretome analysis mut-p53 enhanced secretion of this proangiogenic factor in MDA-MB-231 cells also via miR-30d (Fig. 5f). We evaluated the effect of mut-p53/miR-30d secretome on endothelial cells by performing angiogenesis and vascular permeability assays in vitro using Human Umbilical Vein Endothelial Cells (HUVEC). Preconditioning with MDA-MB-231 CM increased both the number of HUVEC closed loops, and permeability to Dextran of HUVEC monolayers, indicating increased angiogenesis and endothelial leakiness. Both effects were reduced by mut-p53 knockdown and recovered by miR-30d overexpression in mut-p53-KD cells (Fig. 5g, Supplementary Fig. 7e, Fig. 5h).

We then evaluated the effect of the mut-p53/miR-30d secretome on the activation of BJ and Wi-38 normal-like human fibroblasts in wound healing assays. Preconditioning with CM obtained from MDA-MB-231 or MDA-MB-468 cells enhanced fibroblast migration and wound closure; these effects were dampened upon silencing mut-p53 in donor cells, and were recovered by miR-30d overexpression (Fig. 5i; Supplementary Fig. 8a-b). In addition, preconditioning BJ cells with CM from MDA-MB-231 cells grown under hypoxia further enhanced fibroblast migration and wound closure, and these effects were reduced by miR-30d inhibition in donor cells (Supplementary Fig. 8c). Importantly, inhibiting secretion in mut-p53 expressing donor cells by BFA treatment abolished their pro-migratory effect (Supplementary Fig. 8d). We also observed induction of active cancer-associated fibroblasts (CAFs) markers[53] including fibronectin, α-SMA, HIF, and YAP in Wi-38 fibroblasts treated with MDA-MB-231 CM, in a mut-p53/miR-30d-dependent manner (Fig. 5j, Supplementary Fig. 8e-g), indicating that this axis causes functional and metabolic activation of CAFs. Finally, we tested whether this effect of miR-30d required downregulation of DGKZ and VPS26B. Inhibition of miR-30d in MDA-MB-231 cells increased DGKZ and VPS26B expression and concomitantly reduced paracrine stimulation of fibroblast migration. In this context, silencing either DGKZ or VPS26B in MDA-MB-231/miR-30d-dy cells partially restored induction of CAFs migration and wound closure (Supplementary Fig. 8h), confirming that miR-30d promotes CAFs recruitment at least in part via inhibiting these targets.

These results suggest that mut-p53/miR-30d secretome could act locally in a paracrine fashion on tumor and stromal cell populations, both via inducing ECM deposition and remodeling and via signaling by secreted soluble factors.

**miR-30d enhances tumorigenesis and metastasis in vivo.** To investigate the impact of miR-30d-dependent secretome on tumor growth and metastasis in vivo, immunodeficient mice were orthotopically injected with luciferase-expressing MDA-MB-231 cells (MDA-MB-231–LUC), stably transduced with either control or dy-30d construct. Inhibition of miR-30d significantly delayed tumor growth over a 4 weeks period (Fig. 6a).

Of note, immunoblot analysis of primary tumors from engrafted mice highlighted increased expression of DGKZ and VPS26B and reduction of PDIA5, Sec24A, β-COP, and GM130 upon miR-30d inhibition, suggesting decreased secretory trafficking (Supplementary Fig. 9a).

Immunohistochemical analysis of GM130 highlighted enlarged Golgi in primary tumors derived from control mice, while miR-30d inhibition was associated with regular perinuclear Golgi (Fig. 6b). Consistently, analysis of tumor stroma highlighted that miR-30d inhibition strongly decreased ECM deposition/remodeling and CAFs recruitment, as judged by Picro Sirius red and αSMA staining, respectively (Fig. 6c, d). These data suggest a key role of miR-30d in the ability of mut-p53 to support cancer growth by paracrine effects on tumor stroma.

We then evaluated metastatic dissemination by whole-body bioluminescence imaging. We observed a strong reduction of lung metastasis in mice grafted with cells expressing dy-30d as compared with control, even when primary tumors had similar size (Fig. 6e, compare primary tumor and metastasis dimension of CTRL 28 days vs dy-30d 35 days). We also analyzed the impact of miR-30d on metastatic colonization by injecting control- or dy-30d MDA-MB-231–LUC cells intravenously. Total body and ex vivo lung bioluminescence imaging showed that lung colonization was dramatically reduced by inhibiting miR-30d (Fig. 6f–g).

Recent evidence suggests that factors secreted by primary tumors may create a permissive microenvironment for metastasis colonization at secondary sites[1]. Hence, we asked whether miR-30d may contribute to systemic effects of the primary tumor secretome. We pre-conditioned immunocompromised mice with 10 daily intra-peritoneal injections of CM from MDA-MB-231 cells transfected with either IH-30d or control hairpin, prior to engraftment with MDA-MB-231–LUC cells in the mammary fat pad. Since primary tumor growth precludes analysis of secondary lesions, we monitored metastasis growth for 16 days after surgical resection of primary tumors as described[54] (Fig. 6h). Bioluminescence imaging and Ki67 immunohistochemistry highlighted a reduction of metastatic lung colonization upon preconditioning animals with CM derived from IH-30d cells as compared with control (Fig. 6i), with no significant effect on primary tumor growth (Supplementary Fig. 9b). Of note, inspection of metastatic lungs of mice treated with CM from IH-30d cells highlighted reduction of CAFs recruitment, ECM deposition/remodeling and formation of aberrant vessels, as judged by αSMA, Picro Sirius red, and CD31 staining, respectively (Fig. 6j–n). These data suggest that the secretome of mut-p53-expressing cells contributes to generate a permissive microenvironment for metastatic colonization.

We next evaluated the functional impact of the mut-p53/miR-30d axis in human samples. We examined adult primary fibroblasts from two Li-Fraumeni patients, bearing TP53 mutation R248Q, as compared with fibroblasts of two wt-p53 healthy donors. As shown in Fig. 7a, b, mut-p53 expression was associated to increased HIF1α and miR-30d levels, reduced mRNA and protein levels of VPS26B and DGKZ, vesiculated GA (Fig. 7c) and enhanced protein secretion as evaluated by ssGFP (Fig. 7d). Consistently, inhibition of miR-30d dampened secretion in mut-p53 expressing cells (Fig. 7d). These results confirm that the mut-p53/miR-30d axis modifies GA structure and leads to increased secretion also in Li-Fraumeni primary cells, a unique genetic model to study mut-p53 activities in non-neoplastic human tissues[9].

To confirm our findings in established human tumors, miR-30d expression was analyzed by BaseScope in situ hybridization and quantification in tissue sections of invasive ductal breast cancers (n = 12). These were identified as either p53-high cases, bearing hotspot missense TP53 mutations as indicated by Competitive Allele-Specific TaqMan PCR (Supplementary Table 1) and high mut-p53 expression as indicated by immunohistochemistry, or p53-low cases, which displayed no hotspot TP53 mutations and low or null p53 expression. As shown in Fig. 7e, p53-high samples displayed higher miR-30d

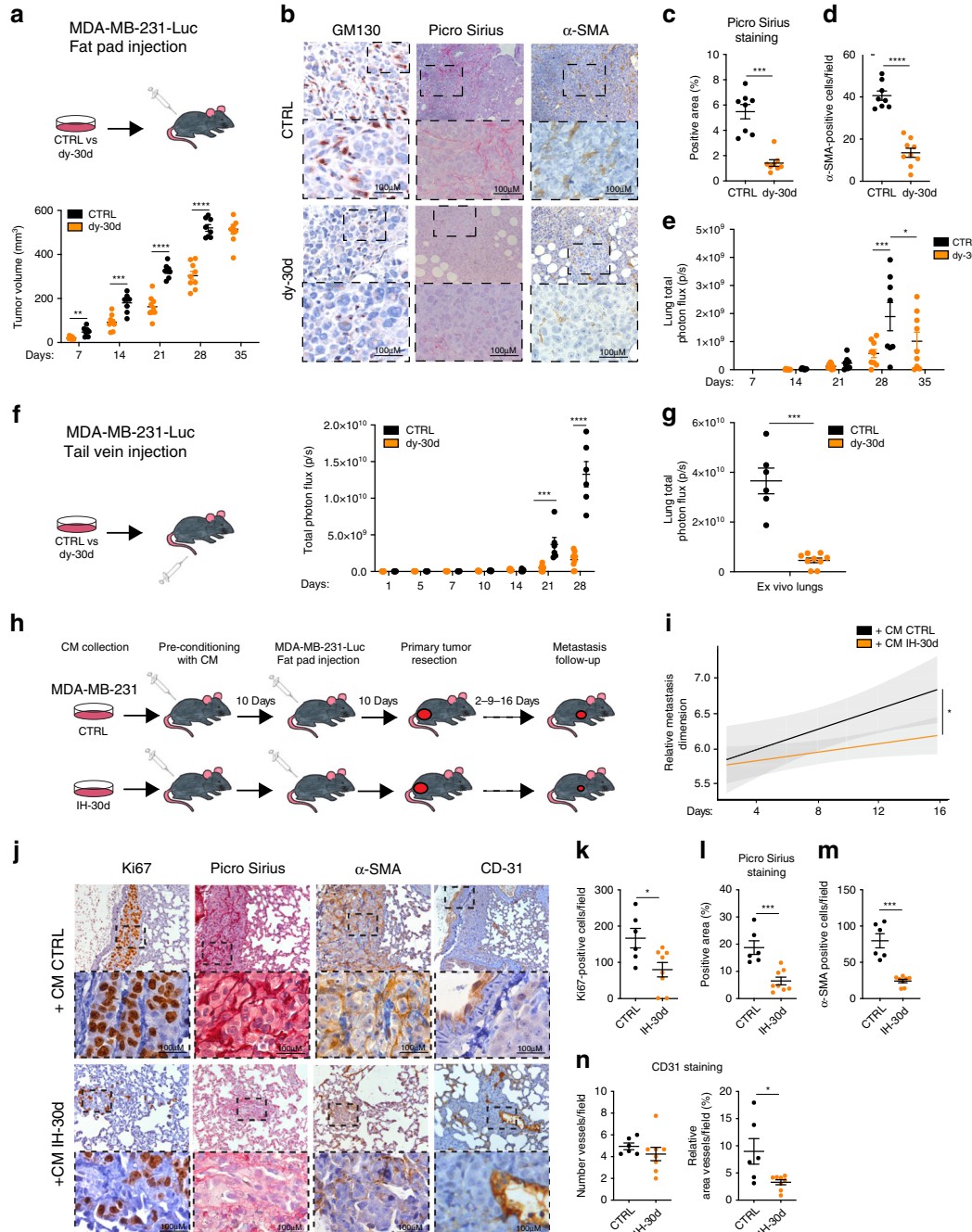

**Fig. 6 miR-30d enhances tumorigenesis and metastasis in vivo. a** Tumor volume measured at the indicated time points after orthotopic xenograft of MDA-MB-231–Luc control (CTRL, n = 8) or miR-30d decoy (dy-30d, n = 9) cells in SCID mice. p value: p = 0.00255667, p = 0.00021304, p = 0.00000072, and p = 0.00000029, respectively, at 7, 14, 21, and 28 days. **b** Representative images of immunohistochemical analyses (IHC) of markers of Golgi (GM130), collagen deposition (Picro Sirius red), and CAFs (α-SMA) in serial sections from primary tumors in (**a**) (CTRL, n = 8; dy-30d, n = 9). Lower panels show higher magnification of upper panels' insets. **c, d** Quantification of immunohistochemistry markers analyzed in (**c**). **e** In vivo luciferase quantification of lung metastases from mice in (**a**). (**a**) and (**b**) report the mean +/− SEM with CTRL (n = 8) and decoy-30d (n = 9) (p value: p = 0.0011 and p = 0.0265, respectively, at 28 and 35 days). **f** Whole-body in vivo luciferase quantification at the indicated time points of control or miR-30d decoy MDA-MB-231–Luc cells injected intravenously in SCID mice (CTRL n = 6, dy-30d n = 9, p = 0.0011134, p = 0.0000023, respectively, at 21 and 28 days). **g** Ex vivo luciferase quantification at 28 days in lungs explanted from mice in (**d**) (CTRL n = 6, dy-30d n = 9, p = 0.0004). **h** Schematic representation of the mouse conditioning experiment. SCID mice were injected with medium conditioned by MDA-MB-231 cells transfected with control or miR-30d inhibitor (IH-30d), before performing orthotopic xenografts. Metastases were evaluated after resection of primary tumors. **i** Confidence lines showing whole-body growth of metastases at the indicated days after surgical removal of primary tumors (Wilcoxon rank-sum test; n = 8 for each group). **j** Representative images of immunohistochemical (IHC) analyses for markers of proliferation (Ki67), collagen deposition (Picro Sirius red), CAFs (α-SMA), and endothelial cells (CD31), in metastases from serial lung sections of mice in (**h**) and (**i**) (n = 8 for each group). Lower panels show magnification of insets in the upper panels. **k–n** Quantification of immunohistochemistry markers reported in (**j**). All graphs report single data points, mean +/− SEM (p value: *p < 0.05, ***p < 0.001, ****p < 0.0001 by two-tailed unpaired Student's t-test). Source data are provided as Source data file.

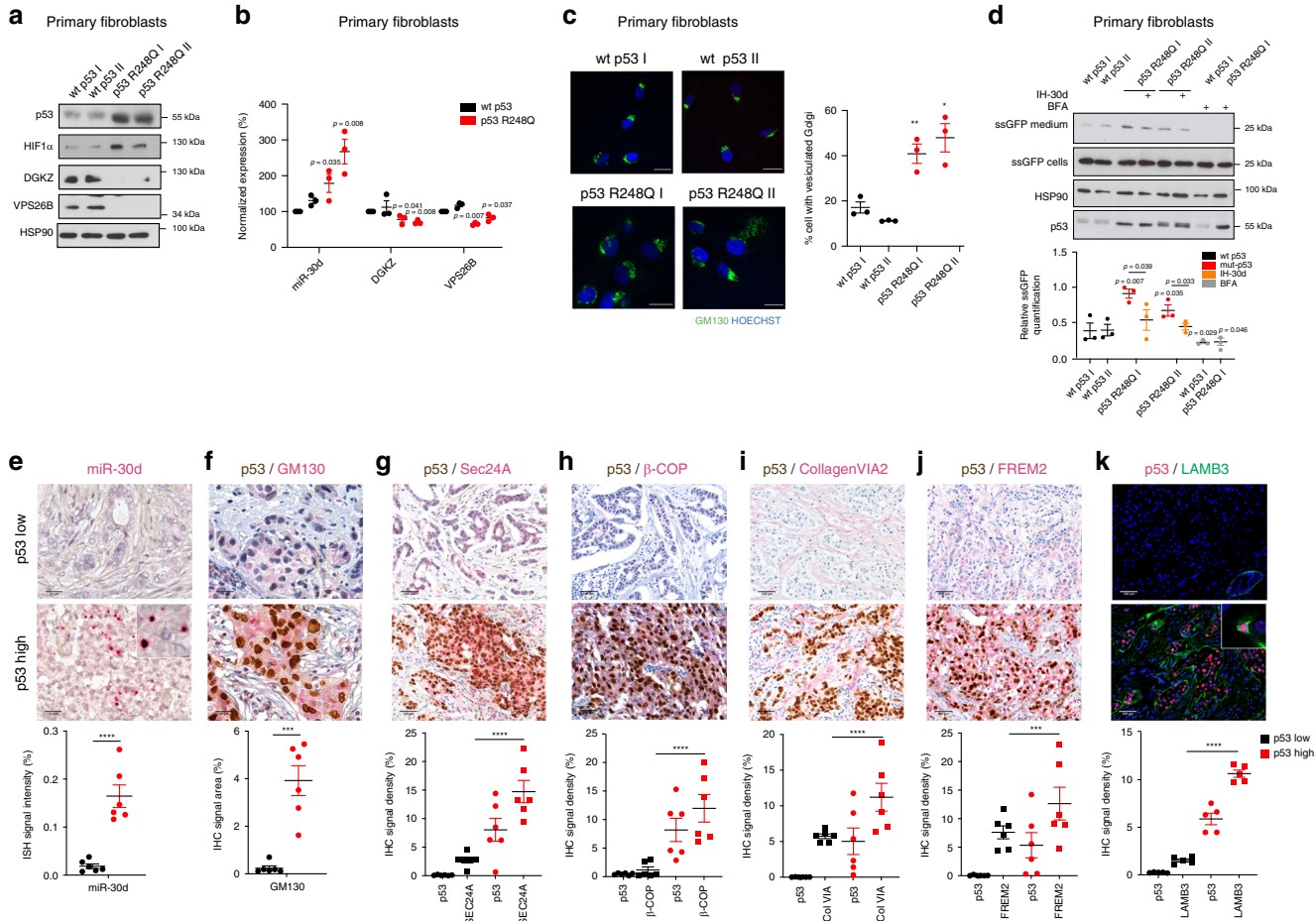

**Fig. 7 Impact of mut-p53/miR-30d axis in clinical samples. a** Analysis of DGKZ, VPS26B, and p53 protein levels in Human Primary Fibroblasts with different p53 status as indicated. HSP90 was used as loading control. The blot is representative of $n = 3$ biological replicates. **b** miR-30d, DGKZ, and VPS26B expression was evaluated in Human Primary Fibroblasts with different p53 status as indicated by RT-qPCR, normalized to U6B and H3 RNA expression levels. Exact $p$ values are indicated in the figure. **c** Immunofluorescence analysis of Golgi apparatus stained with an antibody specific for GM130 in human primary fibroblasts with different *TP53* status as indicated. Right: graph showing the percentage of cells with vesiculated Golgi upon different treatments. **d** Human Primary Fibroblasts with the indicated *TP53* status were transfected with a plasmid encoding green fluorescent protein fused to a secretion signal sequence (ssGFP). Intracellular (cells) and extracellular (medium) GFP protein levels were analyzed upon transfection of miR-30d inhibitor (IH-30d). Forty-eight hours upon silencing, fresh medium was added to the cells and collected after 2 h for analysis. Treatment with BFA 2.5 μM for 2 h was included as control. Densitometric quantification of the ratio of secreted versus intracellular ssGFP is shown in the graph below. All blots are representative of $n = 3$ biological replicates. Exact $p$ values are indicated in the figure. **e–k** Representative images of immunohistochemical (IHC) analysis of breast cancer samples divided on the basis of p53 missense mutation as inferred by p53 staining and allele-specific PCR ($n = 6$ for each condition). Samples were stained with anti-p53 antibody combined with miR-30d in situ hybridization by BaseScope (**e**) or with anti-GM130 (**f**), anti-Sec24A (**g**), anti-β-COP (**h**), Collagen-VI A2 (**i**), FREM2 (**j**), or LAMB3 (**k**) antibodies. The graphs below report the relative quantification of the antigens analyzed. All graphs report single data points, mean +/− SEM; $p$ value (*$p < 0.05$, **$p < 0.01$, ***$p < 0.001$, ****$p < 0.0001$) was calculated by two-tailed unpaired Student's $t$-test. Blots are representative of $n = 3$ biological replicates. Source data are provided as Source data file.

expression, enlarged Golgi judged by GM130 immunostaining (Fig. 7f) and higher Sec24A and β-COP immunostaining, suggesting enhanced secretory trafficking (Fig. 7g, h). In p53-high cases we also observed upregulation of specific components of mut-p53 dependent secretome (Col6A, Frem2, and Lamb3) (Fig. 7i–k).

Altogether these data support a role of the mut-p53/miR-30d axis in promoting cancer progression and metastasis via an altered secretome, exerting paracrine effects on the TME both at primary and distant sites (Fig. 8).

## Discussion

In this work we demonstrate that missense p53 mutants, via transcriptional induction of miR-30d, cause structural modifications of the GA, leading to enhanced secretory trafficking and release of a pro-malignant secretome that modifies the TME. We discovered that mut-p53 induces miR-30d expression by hijacking HIF1α in both hypoxic and normoxic contexts. Activated HIF1α can induce miR-30d also in the absence of mut-p53, implying that hypoxia and other HIF1-inducing stimuli may alter GA structure and secretory trafficking via miR-30d also in tumors lacking mut-p53.

In physiological contexts, increased secretion associates with structural adaptations of trafficking hubs, i.e., enlargement of the ER[6,55] and tubulo-vesiculation of the GA[39]. It is conceivable that oncogenes may exploit these programs to enhance secretion in cancer cells. Indeed, alterations of the GA and expansion of Golgi network have been observed in cancer cells with high tumorigenic potential[7,56], and upregulation of an ER-Golgi trafficking gene signature correlates with breast cancer metastasis[8]. We discovered

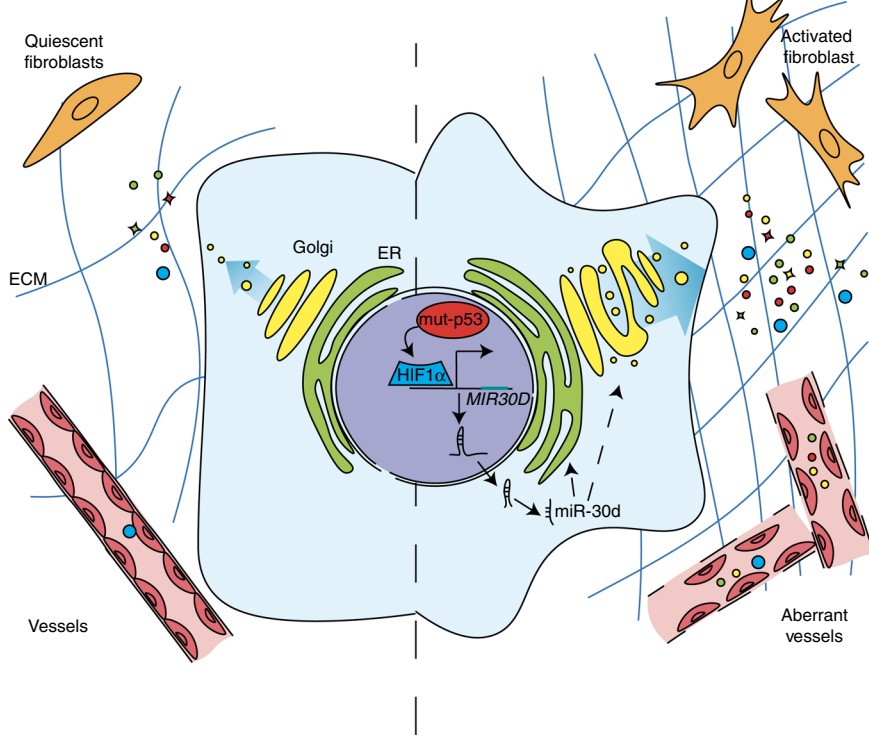

**Fig. 8 Model.** Schematic model of the impact of mut-p53/miR-30d axis on Golgi Apparatus, secretory trafficking and of its paracrine effect on tumor microenvironment.

that the mut-p53/miR-30d axis promotes formation of tubular continuities across Golgi cisternae, resulting in rapid diffusion of cargoes and increased secretion. Mechanistically, we demonstrated that miR-30d affects GA structure in part via direct downregulation of the DGKZ and VPS26B targets, that have been previously linked to modulation of GA structure and secretion. Inhibition of DGKZ kinase by miR-30d leads to accumulation of the polar lipid DAG in Golgi membranes. DAG impacts multiple aspects of Golgi transport and secretory pathway, including GA tubulo-vesiculation[48,57]. Indeed, thanks to its conical shape[49,50] and to its function as second messenger activating PKD/PKCη kinases[58], DAG modulates membrane curvature facilitating vesicular transport and secretion. The second target, VPS26B, is a component of the core retromer complex. Defects in retromer function attenuate retrograde transport from endosomes to the *trans*-Golgi network[46,59]: this may induce GA structural and functional alterations as well as inappropriate sorting and recycling activities of endosomes that may lead to increased secretion, as demonstrated for cathepsin-D precursor[60], and Aβ in Alzheimer's disease[61].

In addition, we observed that miR-30d promotes secretion also by inducing microtubule stability, possibly via indirect effects on kinesins involved in dynamic assembly of Golgi mini-stacks[62,63].

Mut-p53 has been found to upregulate several soluble mediators that promote cancer cell migration and invasion through autocrine or paracrine signaling[15,64]. Our data show that, in addition to regulating the expression of soluble molecules and ECM components in concert with HIF1α[65] and other transcription factors, mut-p53 enhances the whole secretion process, thus enforcing the release of a malignant secretome acting on the TME. Several proteins in the mut-p53 secretome can affect ECM composition and remodeling, contributing to metastatic spread[4]. Accordingly, we observed that the mut-p53/miR-30d-dependent secretome promotes ECM deposition and stiffening, and sustains functional activation of CAFs both at primary and secondary

tumor sites. Of note, both mut-p53 and HIF1α are mechanosensitive factors, activated downstream to actomyosin dynamics induced by a rigid ECM[14,66], and consistently we observed that elevated ECM stiffness increases miR-30d expression and cancer cell secretion. Thus, the mut-p53/HIF1α/miR-30d pathway establishes a vicious cycle that reacts to mechanical cues and, via enhanced secretion, further strengthens these inputs in the TME, increasing tumor aggressiveness, promoting CAF activation and stromal neo-vascularization. These activities contribute to build a supportive tumor stroma at the primary tumor site and at metastatic niches, accelerating the timing of metastasis in vivo. Of note, mouse preconditioning experiments demonstrated that cancer cells deficient for miR-30d expression release a secretome that is unable to support metastatic colonization.

Importantly, we provide evidence that the p53/HIF1α/miR-30d axis contributes to human carcinogenesis, and suggest new potential non-invasive biomarkers and tractable targets to blunt tumor aggressiveness. Interestingly, drugs targeting the GA and secretory pathway are under preclinical study to control tumor growth and dissemination[67]. Our work unveils an oncogene-induced mechanism that offers several actionable targets. Options include interfering with HIF1α function[68,69], blunting mut-p53 gain of function, either through inhibition of the mevalonate/RhoA pathway[14] or other mut-p53 inhibitors (e.g., PRIMA-1Met, Hsp90 inhibitors)[26,70] or directly targeting miR-30d. In sum, alteration of Golgi structure and function by mut-p53 and HIF1α oncogenes confers selective advantages to primary tumor and metastasis, and may be considered as a valuable target for more efficient cancer therapies.

## Methods

**Contact and reagent resource sharing**. Further information and requests for resources and reagents should be directed to and will be fulfilled by the Lead Contact, Giannino del Sal (gdelsal@units.it).

**Human sample collection and patient information**. Human Primary Fibroblasts were obtained by skin biopsy samples collected at The Hospital for Sick Children, Toronto, Ontario, Canada Institutional Research (TP53 mutant, $n = 2$; TP53 wild-type, $n = 2$). Informed consent was obtained from the patient or parent/legal guardian of the patient. Ethics Board approval was obtained for the study, under the study title "Molecular characterization of Li-Fraumeni Syndrome and its variants".

Human breast cancer tissue sections were selected from the archival samples of the Tumor Immunology Laboratory, Human Pathology Section, of the Department of Health Sciences, University of Palermo. The cases were classified according to the World Health Organization classification criteria of the Tumors of the Breast (2013). Samples were collected in accordance with the Helsinki Declaration, and the study was approved by the University of Palermo Ethical Review Board (approval number 09/2018).

**Cell culture**. MDA-MB-231, MDA-MB-468, SUM-159PT, SK-BR-3, HT-29, HEK-293T, and HEK-293GP cells were cultured in DMEM medium supplemented with 10% FBS (Fetal Bovine Serum). H1299, DU 145 and TOV-112D cells were cultured in RPMI medium with 10% FBS. MCF10A cells were maintained in DMEM/Ham's F12 medium in a 1:1 ratio, supplemented with 5% HS (Horse Serum), insulin (10 µg/ml), hydrocortisone (0.5 µg/ml), and epidermal growth factor (EGF, 20 ng/ml). Mahlavu cells were cultured in EMEM medium supplemented with 10% FBS, 1% MEM NEAA (Minimum Essential Medium Non-Essential Amino Acids) and 1% Glutamax.

BJ-EHT-ER-RAS were obtained from R. Agami[71] and cultured in DMEM medium supplemented with 10% FBS. WI-38 cells were cultured in EMEM medium supplemented with 10% FBS and 1% MEM NEAA. HUVEC (Human Umbilical Vein Endothelial Cells) were kindly isolated by Dr. Chiara Agostinis as previously described[72] and provided by Prof. Roberta Bulla, Università degli Studi di Trieste, Trieste, Italy. The cells were seeded in plates coated with fibronectin and maintained in Human Endothelial SFM supplemented with 20 ng/ml bFGF (basic Fibroblast Growth Factor), 10 ng/ml EGF, and 10% FBS. All the experiments with HUVECs were carried out by the fourth cell culture passage.

The LFS fibroblast lines were cultured in AMEM (Wisent #310-022-CL) supplemented with 10% FBS and passaged with 0.05% Trypsin (Wisent #325-542-EL).

All media were supplemented with penicillin and streptomycin (100 IU/mL each). All the cells described were maintained in a 37 °C, 5% $CO_2$ incubator.

Hypoxic conditions were obtained culturing cells in a 37 °C, 5% $CO_2$, 2% $O_2$ incubator balanced with $N_2$, or treating cells with 150 µM of $CoCl_2$ for 16–24 h. To mimic differential substrate stiffness culturing conditions, MDA-MB-231 cells were seeded on top of fibronectin-coated 50 or 0.5 kPa Easy Coat hydrogels (Cell Guidance Systems).

Three-dimensional culture was carried out as previously described[73]. In brief, 24-wells dishes were lined with growth factor-reduced Matrigel, and cells were seeded in DMEM medium supplemented with 1% FBS, 250 ng/ml insulin, 0.5 µg/ml hydrocortisone, and 3% Matrigel for 8 days.

MDA-MB-231 cells, stably expressing TWEEN-EGFP-3′UTR (as control vector) or TWEEN-EGFP-decoy-miR-30d, were maintained with the addition of puromycin to the growth medium (2 µg/ml).

MCF10A, stably expressing pSR-Blast-empty (as a control vector) or miR-Vec-30d were maintained with addition of blasticidin to the growth medium (2 µg/ml).

MCF 10A stably silenced for endogenous wt-p53 (shp53), and overexpressing mut-p53 R175H, mut-p53 R280K, or mut-p53 R273H were selected and maintained with addition of puromycin and blasticidin to the growth medium (2 µg/ml each).

**Transfection and viral transduction**. Cells were transfected when the culture reached 50–80% confluence level. For DNA transfections, the appropriate amount of DNA, depending on the total surface of the culture vessel, was used together with Lipofectamine 2000 or LTX transfection reagents, following manufacturer's instructions; for siRNA/miRNA mimic transfections, cells were transfected with 40 nM siRNA oligonucleotides, 3 nM miRNA mimics, or 20 nM miRNA inhibitor harpin together with Lipofectamine RNAiMax following manufacturer's instructions; as a negative control, the Qiagen AllStars Negative Control siRNA was used. siRNA sequences are listed in Supplementary Table 2.

For retrovirus production, low-confluence HEK-293GP packaging cells were transfected using calcium phosphate with the appropriate plasmids in combination with the pMD2.G packaging vector. For lentivirus production, low-confluence HEK-293T packaging cells were transfected using calcium phosphate with the appropriate plasmids in combination with the pMD2.G and psPAX2 packaging vectors. After 48–72 h, the virus-containing medium was collected and filtered with 0.45 syringe filter to remove cellular debris, and was added to the target cells, which were then selected with puromycin and/or blasticidin, 2 µg/ml each.

**Plasmids**. pRS-shp53 and pRS vectors were kindly provided by R. Agami. miR-Vec constructs were part of the miR-Lib, provided in collaboration by R. Agami[74].

pLPC-ssGFP was obtained fusing the rat FSHb signal peptide[75] upstream of the eGFP gene in the pLPC construct.

GFP-C1-PKCgamma-C1A was a gift from Tobias Meyer (Addgene plasmid # 21205; http://n2t.net/addgene:21205; RRID:Addgene_21205)[51].

psiCHECK2 3′UTR reporter constructs were obtained by cloning each 3′UTR into the psiCHECK2 (Promega) plasmid, downstream of Renilla Luciferase reporter gene, between NotI and XhoI restriction sites. The psiCHECK2 vector also expresses Firefly Luciferase, which is used to normalize for the efficiency of plasmid transfection. The 3′UTRs sequences of AP2A1, DGKZ, IQCG, PPP3CB, and VPS26B were obtained from Ensembl (http://www.ensembl.org/index.html) and UCSC (http://genome.ucsc.edu/cgibin/hgGateway) database and amplified from MDA-MB-231 genomic DNA with AccuPrime™ Taq DNA Polymerase High Fidelity (Invitrogen), following manufacturer's instructions. In the psiCHECK2 DGKZ 3′UTR and psiCHECK2 VPS26B 3′UTR reporters, the miR-30d putative binding sites were mutated by using Quick change II XL Site-Directed Mutagenesis kit (Stratagene, CAT#200521-5), following manufacturer's instructions.

The lentiviral vector pTWEEN 3′UTR EGFP empty was kindly provided by R. De Maria, and the miR-30d decoy was cloned as described by Bonci et al.[37]. N-terminally HA-tagged shRNA-resistant p53 constructs: pMSCV-HA-P53 R175H, P53 R273H, P53 R280K, and pMSCV-HA-P53R were previously described[30].

**RNA extraction and quantitative real-time PCR**. Cells were harvested in Qiazol lysis reagent (Qiagen) for total RNA extraction, and contaminant DNA was removed by DNase treatment. qRT-PCR analyses were carried out on cDNAs retrotranscribed with Quantitect reverse transcription kit (Qiagen), and analyzed genes were amplified using SsoAdvanced™SYBR® Green Master Mix (Bio-Rad) on a CFX96™ Real-Time PCR System (Bio-Rad). For miRNAs and the house-keeping control genes RNU6B and SNORD25 small nuclear RNA, 0.5 µg of total RNA were retrotranscribed and amplified with miScript PCR System (Qiagen) following manufacturer's instructions. The data were analyzed with the Biorad CFX Manager software. Experiments were performed at least three times, and each sample is the average of a technical duplicate. The quantification is based on the $2^{-\Delta\Delta Ct}$ method using the proper housekeeping gene levels as normalization reference. Primers sequences are reported in the Supplementary Table 2.

**Chromatin immunoprecipitation**. Chromatin immunoprecipitation was performed as previously described[30]. Chromatin was immunoprecipitated with the p53 FL-393 (sc-6243, Santa Cruz Biotechnology) antibody. IgGs purified from rabbit serum were used as negative control. Co-immunoprecipitated DNA was analyzed by real-time PCR. Promoter occupancy was calculated as percent of input chromatin immunoprecipitated using the $2^{-\Delta Ct}$ method. Primers sequences are reported in the Supplemental Table 2.

**Microarray data generation**. For microarray analysis of genes regulated by miR-30d, total RNA (2 µg) was isolated from MDA-MB-231 cells expressing miR-30d decoy or control vector. For each experimental condition, three biological replicates were prepared and processed in parallel. RNA concentration, quality and purity were determined using a NanoDrop ND-1000 Spectrophotometer (NanoDrop Technologies Inc.). Synthesis of cDNA and biotinylated cRNA (from 500 ng total RNA) was performed using the Illumina TotalPrep RNA Amplification Kit (Ambion), according to the manufacturer's protocol. Quality assessment and quantification of total RNA and cRNAs were performed with Agilent RNA kits on a Bioanalyzer 2100 System (Agilent). Hybridization of cRNAs (750 ng) was carried out using Illumina Human 48 K gene chips (Human HT-12 v4 Expression Bead-Chip). Array chip washing was performed in High Temp Wash Buffer (Illumina) at 55 °C for 10 min, followed by staining using streptavidin-Cy3 dyes (Amersham Biosciences). Hybridized arrays were stained and scanned in a BeadStation 500 System (Illumina). GenomeStudio Data Analysis Software's Gene Expression Module (GSGX) Version 1.9 was used and cubic spline normalization was applied to the data. The average signal was used for performing the analysis ("AVG_-Signal") using limma in the Biocondoctor. Differentially expressed genes were identified using Significance Analysis of Microarray algorithm coded in the samr R package, estimating the percentage of false-positive predictions (FDR).

**Protein analysis**. To perform western blot analysis, total cell extracts were prepared in lysis buffer (300 mM NaCl, 50 mM Tris-HCl pH 7.5, 1 mM EDTA, 1% NP-40) supplemented with 1 mM PMSF, 5 mM NaF, 1 mM $Na_3VO_4$, and 10 µg/ml CLAP. Protein concentration was determined with Bio-Rad Protein Assay Reagent (#500-0006, Bio-Rad). Lysates were resolved by SDS/PAGE and transferred to nitrocellulose membranes (Millipore). Western blot analysis was performed according to standard procedures.

Co-IP experiments with endogenous proteins were performed using Co-IP buffer (150 mM NaCl, 50 mM Tris-HCl pH 8, 1 mM EDTA, 1% NP-40, 10% Glycerol) supplemented with protease inhibitors as described before. Samples were cleared by centrifugation for 30 min at $13,000 \times g$ at 4 °C and incubated for 16 h at 4 °C with anti-p53 antibody. After 1 h of incubation with protein G-Sepharose (GE Healthcare), immunoprecipitates were washed three times in Co-IP buffer, resuspended in Laemmli sample buffer, and analyzed by western blotting. The antibodies used are listed in the Supplementary Table 3. Anti-mouse and anti-rabbit HRPO-conjugated (Sigma-Aldrich) antibodies were used as secondary

antibodies for Western blot. Bands were quantified by densitometry of autoradiographic films using FIJI software (NIH Image).

**Metabolic labeling.** Cells were cultured in DMEM without L-methionine and L-cysteine for 3 h and then pulsed with 25 μCi/ml of [$^{35}$S] methionine and cysteine (EasyTag™ EXPRESS35S Protein Labeling Mix PerkinElmer) for 1 h. The chase was performed in DMEM containing 0.25 M L-methionine and 0.25 M L-cysteine. For brefeldin-A (BFA) treatment, 2.5 μg/ml of BFA was added to the medium during all phases of the experiments. Medium and cell lysates were collected as described below and analyzed by SDS-PAGE followed by autoradiography. Lanes were quantified calculating the densitometry of autoradiographic film by ImageJ[76], and normalized to the intracellular counterpart.

**Preparation of conditioned medium.** After 6 h from transfection or 24 h from seeding, cells were washed twice with PBS, and grown in serum-free medium for 48 h. Conditioned medium (CM) was collected, centrifuged at $2000 \times g$ for 5 min to remove cells and debris, and filtered through a 0.22 μm syringe filter (EuroClone). If not used immediately, CM was kept a 4 °C for up to 2 days, or stored at −80 °C for up to 1 week.

For analysis of secreted proteins, 1 mL of CM was precipitated by addition of trichloroacetic acid (TCA) (Sigma) to a final concentration of 10% and incubation at 4 °C overnight. The samples were then centrifuged at 17,900 rcf for 15 min at 4 °C, pellets were resuspended in 0.4 ml 100% chilled acetone, and proteins were recovered by centrifugation at 17,900 rcf for 10 min at 4 °C, air-dried at RT for 30 min and dissolved in Laemmli Sample Buffer 2x.

**Secretome analysis.** CM was collected as above, treated with 2% SDS and 0.05 M DTT in 0.1 M Tris-HCl and boiled for 5 min. Total protein concentration in the lysates and the peptide contents in the digests (see below) were assayed using a tryptophan fluorescence based WF-assay in microtiter plate format[77]. The lysates were processed using the MED FASP method with modifications as described[78]. Briefly, proteins were first cleaved overnight by endoproteinase LysC, and subsequently digested with trypsin (enzyme to protein ratio 1:50) for 2 h. Aliquots containing 5 μg total peptide were concentrated to a volume of ~5 μL and stored at −20 °C. Analysis of peptide mixtures were performed using a QExactive HF mass spectrometer (Thermo-Fisher Scientific, Palo Alto). Aliquots containing 2 μg total peptide were chromatographed on a 50 cm column with 75 μm inner diameter packed C$_{18}$ material (Dr. Maisch GmbH, Ammerbuch-Entringen, Germany). Peptide separation was carried out at 300 nL/min for 45 min using a two-step acetonitrile gradient 5–40% over the first 35 min and 40–95% for the following 10 min. The temperature of the column oven was 55 °C. The mass spectrometer operated in data-dependent mode with survey scans acquired at a resolution of 50,000 at $m/z$ 400 (transient time 256 ms). Top 15 most abundant isotope patterns with charge ≥+2 from the survey scan (300–1650 $m/z$) were selected with an isolation window of 1.6 $m/z$ and fragmented by HCD with normalized collision energies of 25. The maximum ion injection times for the survey scan and the MS/MS scans were 20 and 60 ms, respectively. The ion target value for MS1 and MS2 scan modes was set to $3 \times 10^6$ and $10^5$, respectively. The dynamic exclusion was 25 s and 10 ppm. Spectra were searched using MaxQuant software. Proteins were quantified by the 'Total Protein Approach' (TPA)[79].

**Bioinformatics workflow for prediction of putative secreted proteins.** Secretome data were filtered to select putative secreted proteins using a three-step bioinformatics pipeline as described in ref. [80]. First, from 1401 proteins identified, those with predicted signal peptides were extracted using SignalP 4.1[81] with default D-score cut-offs (372 proteins) and UniProt[82] keyword annotation "Signal" (432 proteins), obtaining 450 proteins grouped as "classically" secreted. Of the remaining hits, those annotated with GO[83,84] Cellular Component (GOCC) "extracellular location" were classified as "non-classically secreted proteins I" (557 proteins), discarding those annotated with GOCC "intracellular location". The remaining proteins were analyzed with SecretomeP[85], which predicts secretory proteins following nonclassical, signal peptide-independent mechanisms; proteins with NN-score >0.5 were grouped as "non-classically secreted proteins II" (35 proteins). Merging the classically and non-classically secreted proteins, a list of 1036 proteins were defined as secreted in MDA-MB-231 cell culture media.

**Gene enrichment analysis of the secreted proteins.** For secretome analysis, the differentially secreted proteins were identified performing $t$-test analysis of calculated TPA-values. Then, starting the list of 1036 putative secreted proteins, was filtered including differentially secreted proteins with a Benjamini–Hochberg adjusted $p$ value ≤0.05 in cells depleted of mutant p53 compared with control cells. These proteins were further filtered comparing the obtained list of mut-p53-regulated proteins with the proteins whose basal secretion is restored by miR-30d reintroduction.

**Imaging.** Immunofluorescence staining was performed as previously described[14]. Briefly, cells were fixed in 4% paraformaldehyde for 10 min, washed in PBS, permeabilized with Triton 0.1% for 10 min and blocked in FBS 3% in PBS for 30 min.

Antigen recognition was performed by incubating primary antibody for 1 h at 37 °C and with secondary antibody for 30 min at 37 °C (goat anti-mouse and goat anti-rabbit Alexa Fluor 488, 568, 647, Life Technologies). Nuclei were counter-stained with Hoechst 33342 (Life Technologies). The antibodies used are listed in the Supplementary Table 3.

Golgi morphology was analyzed on ~300 cells for each condition/experiment. To quantify the numbers of Golgi stacks, cells were immunostained with an anti-GM130 antibody and imaged by LSM510 Meta (Zeiss) confocal microscope. For three-dimensional (3D) reconstruction, the Z-stack images of 50 cells/condition were quantified using Volocity image analysis (PerkinElmer).

Analysis of localization of the GFP-C1-PKCgamma-C1A construct, and endogenous PKD-P with the Golgi apparatus (labeled with anti-GM130) was performed calculating the Pearson's $r$ correlation coefficient using the Coloc2 plug-in from the ImageJ suite[76], applying the Costes mask method, in images acquired with Nikon Eclipse C1si confocal microscope. CLEM analysis was performed as described[86,87].

**Proximity ligation assay.** MDA-MB-231 cells were seeded on coverslips 48 h prior to treatment. After hypoxia exposure (2% oxygen for 16 h) or hypoxia-mimetic treatment (150 μM CoCl$_2$ for 16 h), the cells were washed with PBS and fixed in 4% paraformaldehyde for 10 min at room temperature, followed by two washes with PBS. The cells were permeabilized with 0.1% Triton X-100 and washed two additional times with PBS. The PLA was performed using the Duolink In Situ Red Starter Kit Mouse/Rabbit (Sigma) according to the manufacturer's protocol. The following primary antibodies were used: anti-p53 (p53 FL-393, Santa Cruz Bio-technology), diluted 1:100; anti-HIF1α (H1alpha67, Novus Bio), diluted 1:50. The stained coverslips were mounted on slides and visualized with Nikon Eclipse C1si confocal microscope. Representative images are shown for each biological group.

**RUSH system.** Standard procedures using RUSH system were as previously described[41]. MCF10A cells were transiently transfected with plasmids encoding both the hook and the reporter by Lipofectamine LTX Reagent (Life Technologies). 40 μM D-biotin final was introduced at time 0 to release the reporter. Images were acquired using a NIKON Eclipse Ti - Live-cell Imaging system and processed using the FiJi software[76]. Single cells were selected used the built-in ImageJ masking tool and stabilized for instrument-derived shifts employing the Image Stabilizer FiJi plug-in[88]. Total fluorescence intensity of the masked Golgi apparatus area was corrected subtracting the background intensity from an area on identical shape, and the corrected values were normalized on the initial acquisition frame ($t = 0'$).

**Atomic force microscopy analysis.** For ECM stiffness measurement MDA-MB-231 cells were fixed with 4% paraformaldehyde for 20 min. Nuclei were marked via haematoxylin staining, while ECM was visualized using the Picro Sirius Red Stain (Abcam, ab150681). AFM imaging was performed at room temperature on a Smena AFM (NT-MDT Co., Russia) mounted on an inverted fluorescence microscope (Nikon Eclipse Ti-U). For each experimental condition, three biological replicates were prepared and a total of 135 randomly chosen areas were measured and analyzed. The cantilever used was a tip-less probe characterized by a spring constant of about 0.03 nN·nm$^{-1}$ (HQ:CSC38 cantilevers from MikroMasch Co), at the end of which a 18-μm diameter silica bead (Thermo-Fisher Scientific) was glued using UV curable glue (Norland Products Inc.). Force spectroscopy measurements were performed at constant speed (2.5 μm·s$^{-1}$) and triggered to a maximum force applied to the sample of 5 nN. Elastic modulus values, in kPa, were determined by fitting obtained force/displacement curves with a Hertzian model for the tip used taking advantage of the NOVA (NT-MDT Co., Russia) control and analysis software. Statistics and data processing were performed using Igor Pro software (www.wavemetrics.com) and R statistical computing software (www.R-project.org). The significance of the differences in the data was established as equality of probability distributions via the Kolmogorov–Smirnov test.

**Selection of microRNA-30d putative targets.** RNA sequencing data from MDA-MB-231 cells with siRNA of mut-p53 were obtained from the GEO dataset GSE682481, we selected differentially expressed genes as those with a reported Benjamini–Hochberg adjusted $p$ value ≤0.05. Microarray data from MDA-MB-231 cells with shRNA depletion of HIF1α were obtained from the GEO dataset GSE339502, we selected as differentially expressed genes the ones with a false discovery rate (FDR) ≤ 0.05. Microarray data from MDA-MB-231 cells with functional inhibition of miR-30d function were obtained (see above), and we selected as differentially expressed genes the ones with a Benjamini–Hochberg adjusted $p$ value ≤ 0.05. Intersecting the lists of upregulated genes using Venny (Oliveros, J.C. (2007–2015) http://bioinfogp.cnb.csic.es/tools/venny/index.html), 112 genes that were upregulated by mutant p53 depletion, HIF1α depletion, or miR-30d functional inhibition were obtained. This list was then intersected with the 1569 genes that are predicted as miR-30d targets by TargetScan (http://www.targetscan.org/vert_71/4)[89], obtaining the 10 genes shown in Fig. 4a.

**Luciferase assays.** H1299 cells were seeded in 60-mm dishes, and transfected with 2 μg of psiCHECK2 3′UTR reporter vectors. After 24 h the cells were splitted in two plates, and transfected either with 3 nM of miR-30d-mimic or with miR-Negative

Control. Six hours after transfection, medium was changed and 18 h later luciferase activity was measured using the Dual-Luciferase® Reporter Assay System (Promega) on a Promega luminometer. Relative Luciferase Units (RLU) were calculated by normalizing the luciferase units measured for the renilla (*Renilla reniformis*) luciferase on the luciferase units of the firefly (*Photinus phyralis*) luciferase in each sample.

**Capillary-like tubules formation assay**. Tube formation assay was performed as previously described[72]. HUVECs were placed on wells coated with Matrigel (Becton Dickinson) and incubated for 24 h with VEGF (20 ng/mL) or CM from different conditions to allow tube formation. After fixation with 4% paraformaldehyde and staining with Phalloidin-Alexa Fluor 488 (Invitrogen), the number of tubules was counted under a Leica AF6500 microscope using LAS software (Leica).

**Dextran permeability assay**. HUVECs were seeded onto fibronectin-coated transwell inserts (0.4-μm pore size; Corning). When cells were confluent, they were pre-treated for 24 h with CM from different conditions, or for 30 min with TNF-α (100 ng/ml, Invitrogen). FITC-dextran (1 mg/ml, 70 kDa, Sigma) was added to the monolayer (upper chamber) for 30 min. The presence of FITC-dextran in the lower chamber was assayed using Enspire multimode plate reader (PerkinElmer). Fluorescence intensity measurements at 495 nm were expressed as relative permeability by calculating the fold increase over the basal permeability of untreated monolayer (control).

**Wound healing assay**. Equal numbers of BJ-EHT-ER-RAS or WI-38 cells were grown in 6-well or 12-well plates until they reached confluence. Then, cells were scraped with a pipette tip and, after washing with PBS, were incubated with CM from different conditions, or serum-free medium as a control. For each experimental point, two or three scratch were performed; one image of each wounding area was acquired immediately after scratching, and then in the same field after 16 h of migration. The relative wound closure was quantified by measuring the wound area at the time of scratching and at the end point of the experiment using ImageJ[76].

For the experiments performed with WI-38, CM or control medium (DMEM) was diluted 1:1 in the basal medium for culture of WI-38 cells.

**Migration assay**. H1299 cells were grown in 6-well plates in the presence of CM (supplemented with 10% FBS) for 48–72 h. Then $5 \times 10^4$ cells were seeded in 24-well PET inserts (8.0 μm pore size, Falcon) in the presence of CM (supplemented with 1% FBS) in both compartments. After 16 h, cells on the upper part of the membrane were removed with a cotton swab and cells that passed through the filter were fixed in 4% PFA, stained with 0.05% crystal violet and counted.

For the migration assay performed with boiled conditioned medium, CM was first heat-inactivated at 95 °C for 10 min as previously described[90], and then used according to the procedure above described.

**In vivo xenograft experiments**. MDA-MB-231 cells were transduced with a lentiviral vector coding for the firefly (*Photinus phyralis*) luciferase reporter gene, produced in HEK-293T cells. Six- to eight-week-old female NOD/SCID common γ chain knockout (NSG, Charles River) mice were either injected in mammary fat pad with 1 million of cells, or with 100,000 cells intravenously. Orthotopic tumor growth was monitored by caliper measurements and the volume was calculated using the formula: tumor volume (mm$^3$) = $D \times d2/2$, where $D$ and $d$ are the longest and the shortest diameters, respectively.

In vivo imaging was performed at 7, 14, 21, 28, and 35 days after fat pad injection and at days 1, 5, 7, 10, 14, 21, and 28 after i.v. injection. Anesthetized animals (1–3% isoflurane, Merial Italia S.p.A, Italy) were given the substrate D-Luciferin (Biosynth AG, Switzerland) by intraperitoneal injection at 150 mg/kg in PBS (Sigma). Imaging times ranged from 15 s to 5 min, depending on the tumor model and time point. The light emitted from the bioluminescent tumors or metastasis was detected using a cooled charge-coupled device camera mounted on a light-tight specimen box (IVIS Lumina II Imaging System; Caliper Life Sciences, Alameda, CA). Regions of interest from the displayed images were identified around the tumor sites or metastasis regions, such as the lymph node and lungs, and quantified as total photon counts (photon/s) using Living Image® software (Xenogen). For ex vivo imaging, 150 mg/kg of D-Luciferin was injected into the mice just before necropsy. The lungs were excised, placed in a Petri plate and imaged.

Primary tumors were extracted and directly frozen in liquid nitrogen for molecular analyses. Lymph nodes and lungs were excised, formalin-fixed and paraffin-embedded for hematoxylin-eosin staining, Picro Sirius or Ki67, αSMA, and CD31 staining. The antibodies used are listed in the Supplementary Table 3. For the metastasis experiment, mice were treated with preconditioning media for 10 days before fat pad tumor injection. The primary tumor was afterwards removed after 14 days of fat pad injection in anesthetized mice and the lymph nodes and lungs metastases were monitored for 3 weeks.

Procedures involving animals and their care were in conformity with national (D. L. 26/2014 and subsequent implementing circulars) and international (EU Directive 2010/63/EU for animal experiments) laws and policies, and the experimental protocol

(Authorization n. 1143/2015-PR) was approved by the Ethical Committee of the University of Padua (CEASA) and by the Italian Ministry of Health.

**LFS patients, tissues, and sample preparation**. All skin biopsy samples were explanted, established, and cultured by The Centre for Applied Genomics (TCAG) at The Hospital for Sick Children, as described in the protocol below. Briefly, the samples were stored at 4 °C and processed within 2 days, biopsies were dissociated using collagenase, and then treated with Trypsin EDTA; the obtained cells were cultured until the cell line could be expanded, frozen, and stored in liquid nitrogen until required. The cell lines used were derived from 5 or fewer passages.

**Analysis of p53 status**. Genomic DNA was isolated from FFPE breast cancer tissue samples using the QIAamp DNA FFPE Tissue Kit (Qiagen).

The TP53 mutational status was assessed by TaqMan mutation Detection Assay (Life Technologies), based on Competitive Allele-Specific TaqMan PCR (CastPCR) technology to detect the following TP53 mutations: (c.524G>A/p.R175H, c.535C>T/p.H179Y c.742C>T/p.R248W, c.659A>/p.Y220C, c.818G>T/p.R273L, c.488A>G/p.Y163C, c.711G>T/p.M273I, c.517G>T/p.V173L, c.818G>A/p.R273H, c.743G>A/p.R248Q), according to the manufacturer's instructions.

**Immunohistochemistry analyses**. Immunohistochemistry was carried out on FFPE mouse and human tissue sections. Briefly, sections 2.5/3-micron-thick were cut from paraffin blocks, dried, de-waxed, and rehydrated. The antigen unmasking technique was performed using Target Retrieval Solutions pH 6 and pH 9 in a PT Link Dako pre-treatment module at 98 °C for 30 min. Subsequently, the sections were brought to room temperature and washed in PBS. After neutralization of the endogenous peroxidase with 3% $H_2O_2$ and Fc blocking by a specific protein block (Novocastra, UK), double-marker immunohistochemistry was carried out by incubation overnight at 4 °C with a cocktail of two primary antibodies with different made (i.e., mouse and rabbit).

The mouse tissue samples were incubated with the following primary antibodies: Ki67 (Abcam ab16667), αSMA (Abcam ab5694), and CD31 (Abcam ab124432), and with the Dako envision HRP-labeled polymer anti-rabbit secondary (ThermoScientific). Staining was revealed using DAB Quanto Kit (ThermoScientific). The Picro Sirius Red staining was performed as indicated by manufacturer (Picro Sirius Red Stain Kit ab150681). The slides were counterstained with Harris hematoxylin (Novocastra), and images acquired with ×40 objective in the DM4000B microscope (Leica) using the software Leica Application Suite 4.12.

Human tissue samples were incubated with the following primary antibodies: mouse Monoclonal p53 (clone DO-7; 1:50 pH 6; Novocastra), rabbit Polyclonal Sec24a (1:50 pH 6; Abcam), rabbit Polyclonal COL6A2 (1:100 pH 6; Invitrogen), rabbit Polyclonal FREM2 (1:200 pH 6; Abcam). Staining was revealed using MACH2 Double Stain detection kit (Biocare Medical), DAB (3,3'-diaminobenzidine) and Vulcan Fast Red as substrate chromogens. The slides were counterstained with Harris hematoxylin (Novocastra).

For double-marker immunofluorescence, sections were incubated with the following primary antibodies: mouse Monoclonal p53 and mouse Monoclonal Laminin beta-3 (clone CL3363; 1:200 pH 6; Invitrogen). The binding of the primary antibodies to their respective antigenic substrates was revealed by Opal Multiplex IHC kit, which allowed for combined immunostainings using antibodies with a same made through tyramide signal amplification. After deparaffinization, antigen retrieval was performed using microwave heating in pH 6 or pH 9 buffer and the first primary antibody was incubated. Immunofluorescence labeling was achieved by incubating with a specific secondary antibody followed by the addition of one selected Opal fluorophore and microwave treatment in pH 6 buffer. The same procedure was repeated for the second primary antibody using a different Opal fluorophore and DAPI nuclear pigment.

Human MiR30d probe hybridization (Hs-pre-MIR30D-1zz-st; Gene ID 407033; RefSeq accession NR_029599.1; Cod. 720351; Advanced Cell Diagnostic) was performed using BaseScope Detection kit (Advanced Cell Diagnostic) in accordance with the manufacturer's protocol[91] adopting an extended 1-h incubation in Amp 5 and 30-min incubation in Amp 6 buffers.

All the sections were analyzed under a Zeiss AXIOScope.A1 optical microscope (Zeiss, Germany) and microphotographs were collected using a Zeiss Axiocam 503 Color digital camera using the Zen 2.0 imaging software.

For quantification analyses of Ki67 and αSMA staining in metastatic lung samples from mice, positive cells were detected and counted. The Picro Sirius Red Stain positive areas were quantified calculating the labeled area over a fixed threshold normalized to total area by the ImageJ suite[76]. The CD31 staining was analyzed evaluating the number of vessels and calculating the vessel area normalized to total area by the Fiji ImageJ suite[76]. All the analyses were performed in three different fields/mouse in at least 6 mice/condition.

In human breast cancer tissue section analisys, to reduce the presence of stochastic noise for quantification analyses of images with high magnification factor, we applied the methodology described in ref. [92] that maintains details of the finest components. A further pre-processing step consists in the normalization of images to have the same average and standard deviation of their RGB color values: all images result comparable and do not require to be handled individually. The final segmentation was obtained by applying the so-called "‡ trous" wavelet

transform that retains the same resolution as the original photomicrograph[93]. Quantitative in situ hybridization analysis for miR-30d in tumor foci was performed by calculating the average percentage of positive signals in five non-overlapping fields at high-power magnification (×40) using the Positive Pixel Count v9 Leica Software Image Analysis.

**Collection and processing of breast cancer gene expression data**. Breast cancer gene expression data have been obtained from a collection of 4640 samples from 27 major datasets comprising microarray data of breast tumors annotated with pathological information and clinical outcome. All data were measured on Affymetrix arrays and have been downloaded from NCBI Gene Expression Omnibus (GEO, http://www.ncbi.nlm.nih.gov/geo/) and EMBL-EBI ArrayExpress (http://www.ebi.ac.uk/arrayexpress/). Prior to analysis, all datasets have been re-organized as described in ref. [94]; gene expression levels were normalized and quantified as previously described[95]. The type and content of pathological and clinical annotations have been standardized, among the various datasets, as described in ref. [96]. This resulted in a compendium (meta-data set) comprising 3661 unique samples from 25 independent cohorts. Average signature expression has been calculated as the standardized average expression of all signature genes in sample subgroups (e.g., p53 status). The values shown in the bar graph are thus adimensional. All gene expression analyses were performed in R (version 3.5.1).

**Statistical analyses and reproducibility**. All the experiments are representative of at least three independent replicates. All graphs represent single data point mean ± SEM. Statistical tests were performed using GraphPad Prism8. $P$ values were obtained using two-tailed Student's unpaired parametric $t$-test. Blot and micrographs are representative of three independent experiments, experiments for which we showed representative images were performed with similar results at least three independent times.

The commercial reagents, experimental models and software used in the present study are listed in Supplementary Table 4.

**Reporting summary**. Further information on research design is available in the Nature Research Reporting Summary linked to this article.

## Data availability

The authors declare that the data supporting findings of this study are available within the paper and its supplementary information files. Array data that support the findings of this study have been deposited to GEO, accession number GSE133410. Lists of differentially secreted proteins by mut-p53/miR-30d axis in MDA-MB-231 cells are reported in Supplementary Data file 2. Public available data used in this were obtained from: Ensembl (http://www.ensembl.org/index.html); UCSC (http://genome.ucsc.edu/cgibin/ hgGateway); DAVID Bioinformatics Resources 6.8 (https://david.ncifcrf.gov/); GSEA Gene Set Enrichment Analysis (https://www.gsea-msigdb.org/gsea/index.jsp); TargetScan (www.targetscan.org); GOCC (http://geneontology.org/); SecretomeP 2.0 (http://www.cbs.dtu.dk/services/SecretomeP/); SignalIP (http://www.cbs.dtu.dk/services/SignalP/); UniProt (https://www.uniprot.org/); Molecular Taxonomy of Breast Cancer International Consortium, METABRIC[97]; The Cancer Genome Atlas (TCGA) breast cancer dataset (https://www.cancer.gov/about-nci/organization/ccg/research/structural-genomics/tcga). Source data are provided with this paper.

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

## Acknowledgements

We thank A. Testa for discussion and proofreading the manuscript and G. Pastore for technical support. We would like to thank M. Foiani and G. Scita (Milan, IFOM institute) for suggestions, Y. Ciani for advices on statistical data, C. Valenti for IHC quantification of BC samples, and D. Vacca for analyzing TP53 status in BC tissues. We acknowledge support by the Fondazione AIRC IG grant 22174 and "5 per mille Special Programs" grants ID 22759 and ID 10016, Regione Autonoma Friuli Venezia Giulia (Contributo ex art. 15 L.R. 17/2014) to G.D.S., the AIRC IG-21354 to A.R., and AIRC IG-14173 to L.C. We

acknowledge support by the Canadian Institutes for Health Research and the Terry Fox Research Institute (with funds from the Terry Fox Foundation) to D.M. We thank the Confocal microscopy facility of the University of Trieste. We would like to remember Dr. Guido Perelli-Rocco, president of the Fondazione AIRC (Italian Association for Cancer Research) Friuli Venezia Giulia for his tireless and passionate support to our work.

## Author contributions

V.C. designed and performed the majority of the experiments, analyzed the data, prepared the Figures and wrote the paper. L.B., M.F., and E.C. performed experiments. G.B. and A.M. performed the CLEM experiment. R.S. and A.R. performed mice experiments. V.C. and A.B. performed the miRNAs screening. V.Can. and C.T. provided the human tumor samples and human sample immunohistochemistry. J.R.W. performed the mass spectrometry analysis. G.D.S. supervised the project, designed experiments and wrote the paper. L.U.S. and D.S. performed the AFM measurement. F.B. performed the Tube Formation Assays. S.P. performed the MicroArray data analysis. J.L., N.A., L.B., and D. M. provided human primary fibroblasts. V.C., S.B., and M.F. performed bioinformatic analysis. F.M. and L.C. supervised specific experiments and edited the paper. G.D.S. supervised the project, designed experiments, and wrote the paper.

## Competing interests

The authors declare no competing interests.
