## [Peer Review File · Nature Communications]

Reviewers' Comments:

Reviewer #1:

Remarks to the Author:

Capaci et al report that mutant p53 (mut-p53) upregulates the expression of miR-30d in cooperation with HIF-1a. This leads to structural changes in the Golgi apparatus, which drive augmented protein secretion. They further identify two mRNAs that are targets for direct inhibition by miR-30d and confirm their importance in driving the changes in the secretory machinery. Additionally, they identify a panel of proteins whose secretion is significantly elevated as a consequence of mut-p53/miR-30d action, and show that the altered secretome is responsible for augmenting the migratory behavior of cancer cells and of cancer-associated fibroblasts, as well as for increased angiogenesis. Through in vivo experiments in mice, they provide compelling evidence that the mut-p53/miR-30d axis promotes metastatic spread, by acting both at the primary tumor site and at the premetastatic niche. Lastly, they provide data from the analysis of human patient samples which support their conclusions.

Overall, this is a very interesting paper, describing a novel gain-of-function activity of mut-p53 with high potential clinical relevance. The work is comprehensive and is well performed. Publication is recommended after the authors address the points listed below.

Major comments:

1. The paper gives the impression that ALL the effects of mut-p53 on the secretory machinery are mediated by miR-30d. Is this indeed so, or is this an oversimplification of a more complex picture? Specifically, does mut-p53 regulate directly the expression of any genes that encode proteins involved in the secretory machinery? The authors should compare by RNA-seq the transcriptome of miR-30d depleted (dy-30d) cells with and without depletion of mut-p53, and determine whether any secretory machinery-related genes are still regulated by mut-p53 even when miR-30d is absent. The answer to this question is important in order to place the paper's findings within a more complete picture.
2. Fig. 1f: This result is a bit puzzling. Is it not surprising that HIF1a protein levels are not higher in the hypoxic conditions? Actually, if one compares them to the p53 band intensity in the input lanes, they even appear to be lower than in 20%! Moreover, the interaction between mut-p53 and HIF1a does not appear to be increased in 2% when compared to 20%. From this analysis, and considering the model in Fig. 1h, it is unclear what is driving the increased expression of miR-30d under hypoxic conditions. Is it because HIF-1a binds better to the miR-30d chromatin under hypoxia? This should be tested by performing a HIF1a ChIP under conditions similar to those used in Fig. 1g. Moreover, the model in Fig. 1h predicts that more mut-p53 should be bound to the miR-30d locus under hypoxia than in normoxia. This should be tested by comparing the p53 ChIP as in Fig. 1g between 20% and 2%.
3. Fig. 5f: the effect of miR-30d on VEGFA secretion by mut-p53-depleted cells is rather marginal, suggesting that most (90%?) of the effect of mut-p53 is not due to miR-30d. The authors should quantify the band intensities. As a matter of fact, Fig. 5h also shows only a very partial phenotypic rescue by miR-30d, which may be consistent with the limited effect of miR-30d on VEGFA secretion. Together, this suggests that much of the effect of mut-p53 on VEGFA secretion and activity is NOT via miR-30d, unlike what one may conclude from reading the text. The authors should discuss this issue.

Minor comments:

1. Fig. 1f: the IgG control lane in the 20% panel seems to have a bubble, making the expected negative result less convincing. It will be helpful if the authors can provide a cleaner figure, showing clearly that there is no HIF1a band in the IgG control.
2. Fig. 3a: The PDI staining in the mut-p53 R280K cells look very different from that in the miR-30d cells. The authors need to comment on this, or provide a more representative image if these differences are not representative.
3. Supplemental Fig. 5c suggests that miR-30d overexpression causes downregulation of mut-p53,

but this is not seen in Supplemental Fig. 5e. Please explain.

4. Line 181: 3e should be 2e.

5. Line 333; 5a should be 5b.

6. Line 369: 'ad' should be 'and'

Reviewer #2:

Remarks to the Author:

NCOMMS-19-40638

Mutant p53 induces Golgi tubulo-vesiculation driving a prometastatic secretome

Valeria Capaci, Lorenzo Bascetta, Marco Fantuz, Galina Beznoussenko, Roberta Sommaggio, Andrea Bisso, Alexandre Mironov, Valeria Cancila, Jacek Wisniewski, Luisa Ulloa Severino, Denis Scaini, Fleur Bossi, Jodi Lees, Noa Alon, Ledia Brunga, David Malkin, silvano Piazza, Licio Collavin, Antonio Rosato, Silvio Bicciato, CLAUDIO TRIPODO, Fiamma Mantovani, and Giannino Del Sal

While p53 is one of the most critical tumor suppressor, mutant p53 serves as an oncoprotein in Li-Fraumeni cancer predisposition syndrome. In this study, Dr. Del Sal and colleagues determined the role of mut-p53 in tissue culture cells, animal models and human tissues. They found that p53 mutants interact with the hypoxia responsive factor HIF1a and induce the expression of miR-30d, which reduces the DGKZ and VPS26B protein levels and causes tubulo-vesiculation of the Golgi apparatus. Subsequently, this enhances vesicular trafficking and secretion and promotes secretion of growth factors such as VEGF and ECM proteins. Thus, the authors have discovered a novel mut-p53/HIF1a/miR-30d axis that impacts synthesis, trafficking, and secretion of a number of proteins essential for tumor growth and metastatic colonization.

Overall the topic is novel and significant, and a large amount of data was provided, covering a broad range of aspects of cancer cell biology. However, some of the conclusions are based on weak results or speculations. There is also some critical information missing in the manuscript.

Specific points:

1. The interaction of mut-p53 and HIF1 is new, but some critical questions are not addressed.

1) Figure 1e, can HIF over expression rescue the sip53 effect?

2) Figure 1f, HIF1 activity is known to be higher at low oxygen level, if the authors' hypothesis is that mut-p53 acts as a partner of HIF1a, why the interaction is lower at low oxygen?

2. The manuscript did not provide an adequate explanation of how the Golgi becomes vesiculated.

1) Fig. 3a, the shp53+p53 R280K cells have a different phenotype from miR-30d cells for beta-COP and GM130. What's the explanation?

2) The authors identified two proteins, DGKZ and VPS26B, that may affect Golgi morphology and protein trafficking, but it is not clear how. DGKZ is an enzyme that affects DAG to PA conversion, there is no mechanism for how it affects Golgi structure and function. VPS26B is involved in retrograde trafficking, no mechanism was provided whether it also affects anterograde trafficking and secretion.

3) The authors should at least test a few key Golgi structural and trafficking proteins (e.g. Sec24) by Western blot in cells of the 4 different treatments as in Figure 3A and see which protein is affected. Results from tissue culture cells should be confirmed in mouse and human tissues.

3. It is not clear how Golgi vesiculation enhances protein trafficking and secretion.

1) It is not clear how Golgi vesiculation enhances protein trafficking or secretion. The authors speculated that microtubule destabilization or degradation may cause Golgi fragmentation, but this is inconsistent with the fact that destabilization of microtubules slows down protein trafficking, instead of accelerating it.

- 2) Changes in trafficking may not always impact secretion depending on the time when they are determined. It is not described how long the cells were incubated before ssGFP was collected in the medium, but it was shown that at this time most ssGFP was still inside the cells. Did the authors include cycloheximide in this experiment? Since most protein trafficking from the ER to the plasma membrane takes about 2 hours, where is ssGFP stuck in cells? How long were the cell incubated when the secretome was analyzed?
- 3) In general, secretory proteins are either degraded by ERAD if misfolded or secreted. Their trafficking speed may be modulated by the trafficking machineries, but they are normally not held in the Golgi. The comparison of secretome with the transcriptome from a different study in Figure 2h is not valid as these two studies may have different experimental conditions or settings.
- 4) Is accelerated trafficking and enhanced secretion selective for some proteins but not others? This should be confirmed with a few target proteins.
- 5) Related to above, it is necessary to figure out whether the reduced secretion of specific proteins is because of reduced synthesis or trafficking.
- 6) Figure 3d, ManII is a Golgi marker and so can only be used to measure ER-to-Golgi trafficking, not intra-Golgi trafficking related to Golgi vesiculation. In this figure, the Golgi does not seem to be vesiculated in the miR-30d cells. There is also no quantitation of the movie.
- 7) Movie 2 is not convincing. Since E-Cad is N-glycosylated, the authors could use EndoH treatment and bandshift to indicate the trafficking speed.
- 8) Figure 4g, the pPKD pattern is odd, its localization to the Golgi is not convincing.

4. Some critical controls or details are missing.

- 1) miR-30d was previously identified as a negative regulator of p53 (Kumar et al., Oncogene, 2011). This was not discussed in this study.
- 2) On the other hand, several known microRNAs regulated by p53 such as miR-34 were not included in the screening.
- 3) In addition to the p53 mutants, WT p53 should be tested in parallel in some of the critical experiments.
- 4) The authors collected conditioned medium from MDA-MB-231 and H1299 cells, but did not mention how long the cells were incubated before the conditioned medium was collected. In addition, MDA-MB-231 cells were cultured in DMEM while H1299 cells were cultured in RPMI1640; the two different media have different compositions.
- 5) Molecular weight markers are missing for some gels, scale bars are missing in some micrographs.
- 6) For the proteomic and bioinformatic studies it would be helpful to indicate what proteins/genes are included in which categories and how much they have changed under different conditions. Some of the tables include the protein/gene names but did not show which categories they belong to.

Reviewer #3:

Remarks to the Author:

This is an interesting manuscript that documents a novel gain of function activity of mutant p53 proteins in breast cancer. The authors intriguingly show that the transcriptional axis mutp53/HIF1a promotes aberrant activation of miR-30d that provokes the release of a pro-metastatic secretome. The listed below comments need to be fully addressed to render the manuscript more logically organized and more balanced between in vitro and vivo reported findings.

Specific comments:

1. Most of the reported in vivo experiments refer to the mutp53-miR30d axis lacking any evidence that includes HIF1a. This is not consistent with the transcriptional regulatory role that mutp53(HIF1a exerts on miR-30d expression; thereby providing a set of data lacking the main transcriptional and functional player such as HIF1a.
2. Do breast cancer patients carrying TP53 mutations in Metabric and TGCA or other datasets

exhibit higher levels of miR-30d? What about the reported and investigated miR-30d targets? Is their expression anti-correlated to miR-30d?

3. TP53 sequencing and LOH analyses of the 12 analyzed human breast cancers is required to confirm that they are representative of specific mutant p53 gain of function tumors. P53 immunostaining isn't sufficient for making the conclusion proposed by the authors.

4. It appears rather unclear why adult fibroblasts from Li-Fraumeni represent a unique genetic model to study the impact of mutp53/miR-30d axis on breast tumorigenesis. If any, it's difficult to believe that fibroblasts can recapitulate the complexity of established breast cancer tumors. One intriguing possibility could be that of IHC of breast cancer tissues from LFS patients.

5. Does HIF1a depletion affect the mutp53-miR30d dependent secretome? While the authors initially showed that miR30d expression is controlled by the transcriptional crosstalk mutp53/HIF1a, most of the in vitro and vivo biological experiments have been performed lacking any evidence on the role of HIF1a in mutp53-miR30d axis.

6. How the authors explain that mutap53/HIF1a bind miR30d regulatory regions in both normoxic and hypoxic conditions? The described transcriptional complex is active in both conditions? What about the analysis of markers of chromatin modifications or the recruitment of acetylase or deacetylase in both conditions to address this issue.

7. Recruitment of HIF1A on miR-30d promoter needs to be analyzed as well, to conclude that mut-p53 acts as a partner of HIF1a, leading to transcriptional upregulation of miR-30d expression.

8. Effect on secretion after HIF1A stabilization in presence/absence of miR-30d inhibition needs to be analyzed to conclude that the mutp53/HIF1A complex impinges on secretion through miR-30d induction and activity.

9. Golgi vesiculation has been mainly analyzed in non-transformed cells throughout the manuscript with the majority of results obtained in MCF10A cells and fibroblasts. The relevance of the identified network in cancer cells could be strengthened by the inclusion of additional analyses performed in cancer cells after modulation of the key players of the identified network.

Minor comments:

1. Figure 2, was GSEA performed only considering the protein secretion pathway? Information provided in the results/legend for GSEA analysis is incomplete (q-val? Enrichment?).

2. Does miR-30d impact on proliferation? What about mutp53? If this is the case, secretion should be analyzed also after additional cell cycle arresting stimuli.

3. Fig2g, the description is not clear, please include in the matrix the control samples siSCR and miR-CTR

4. Fig2. Venn diagram of proteins controlled by mp53 and by miR-30 would help in understanding results

Here below you will find the point by point responses to the Reviewers' comments.

--

Reviewer #1 (Remarks to the Author):

Capaci et al. report that mutant p53 (mut-p53) upregulates the expression of miR-30d in cooperation with HIF-1a. This leads to structural changes in the Golgi apparatus, which drive augmented protein secretion. They further identify two mRNAs that are targets for direct inhibition by miR-30d and confirm their importance in driving the changes in the secretory machinery. Additionally, they identify a panel of proteins whose secretion is significantly elevated as a consequence of mut-p53/miR-30d action, and show that the altered secretome is responsible for augmenting the migratory behavior of cancer cells and of cancer-associated fibroblasts, as well as for increased angiogenesis. Through in vivo experiments in mice, they provide compelling evidence that the mut-p53/miR-30d axis promotes metastatic spread, by acting both at the primary tumor site and at the premetastatic niche. Lastly, they provide data from the analysis of human patient samples which support their conclusions.

Overall, this is a very interesting paper, describing a novel gain-of-function activity of mut-p53 with high potential clinical relevance. The work is comprehensive and is well performed. Publication is recommended after the authors address the points listed below.

Major comments:

1. The paper gives the impression that ALL the effects of mut-p53 on the secretory machinery are mediated by miR-30d. Is this indeed so, or is this an oversimplification of a more complex picture? Specifically, does mut-p53 regulate directly the expression of any genes that encode proteins involved in the secretory machinery? The authors should compare by RNA-seq the transcriptome of miR-30d depleted (dy-30d) cells with and without depletion of mut-p53, and determine whether any secretory machinery-related genes are still regulated by mut-p53 even when miR-30d is absent. The answer to this question is important in order to place the paper's findings within a more complete picture.

Mut-p53 is known to impact a vast array of cellular processes through regulating, either directly or indirectly, a large number of coding and non-coding genes. It is thus conceivable that some of these may affect cell secretion. To answer the reviewer's questions, we employed two different approaches.

Performing and analyzing a new RNA-seq experiment has proved unfeasible due to limitations for activities both in the lab and outsourcing services (sequencing), associated to COVID-19 lockdown. Therefore, to highlight putative direct target genes of mut-p53 that are involved in the secretory pathway, we inspected -omic data previously generated by our group, namely mut-p53 CHIP-seq and RNA-seq in MDA-MB-231 cells (Walerych et al., 2016). Specifically, we retrospectively searched mut-p53 CHIP-seq data for peaks indicating direct binding of mut-p53 to promoters within a curated list of 96 genes involved in Protein Secretion Pathway obtained from GSEA gene set collection (HALLMARK_PROTEIN_SECRETION). Interestingly, this analysis highlighted that mut-p53 specifically binds to chromatin regions upstream of 21 of the 96 genes investigated. Consistently, mRNA levels of 14 of these 96 genes were also found modulated in RNA-seq analysis performed in MDA-MB-231 upon silencing mut-p53 (see table below), suggesting that these might represent true direct targets of mut-p53.

14 common elements in "Hallmark protein", "CHIP-seq" and "siP53 RNAseq":

AP2S1
AP3S1
ATP7A
COPB1
ERGIC3
GBF1
GOLGA4
LAMP2
NAPA
SEC22B
SGMS1
STAM
TMED2
YKT6

Of these 14 genes, 4 (LAMP2, COPB1, STAM and AP2S1) are also found regulated by miR-30d in the same cells line, by transcriptomic analysis shown in this manuscript.

Altogether, these analyses suggest that mut-p53 may indeed directly regulate genes involved in the secretory process independently of miR-30d. However, validation by RT-qPCR and ChIP-PCR would be required in order to confirm this hypothesis. We believe that this analysis goes beyond the scope of our work; for this reason, and considering the allowed text length, we did not include these data in the manuscript.

Nonetheless, we decided to address the important question raised by the reviewer, as to whether all the effects of mut-p53 on secretion are mediated by miR-30d. In particular, we decided to estimate whether mut-p53 still regulates the process of secretion even when miR-30d is inhibited.

To this aim we evaluated the effect of silencing mut-p53 on ssGFP secretion in cells treated with miR-30d inhibitor. This experiment highlighted that inhibition of miR-30d caused a significant reduction of ssGFP secretion (about 50%), however silencing of mut-p53 in this context did not further affect ssGFP secretion. This suggests that, at least in this experimental setting, miR-30d acts as a major mediator of the impact of mut-p53 on secretion. These data are shown in new Supplementary Fig. 4g, and discussed in the Results section at pag. 6.

2. Fig. 1f: This result is a bit puzzling. Is it not surprising that HIF1 α protein levels are not higher in the hypoxic conditions? Actually, if one compares them to the p53 band intensity in the input lanes, they even appear to be lower than in 20%!

We agree with the comments. Indeed, in the original Fig. 1f, panels showing co-IP under normoxic (20% pO₂) or hypoxic (2% pO₂) conditions were derived from independent experiments, which cannot be directly compared. In the new Fig. 1e we now provide a WB comparing equal amounts of protein lysates from MDA-MB-231 cells grown under different oxygen pressure, which clearly shows that HIF1 α protein levels are increased under hypoxic conditions.

Moreover, the interaction between mut-p53 and HIF1 α does not appear to be increased in 2% when compared to 20%.

To compare the interaction between endogenous mut-p53 and HIF1 α proteins under normoxic and hypoxic conditions within the same experiment, we performed both co-IP and proximity ligation assays (PLA) in cells grown under normoxic (20% pO₂) or hypoxic (2% pO₂) conditions, as well as upon hypoxia-mimetic treatment with CoCl₂. These new experiments, shown in new Figure 1f and Supplementary Figure 1k, confirmed our previous finding that the interaction occurs in both normoxic and hypoxic conditions, and further highlighted that the binding of mut-p53 with HIF1 α

increases of more than two-fold under low oxygen pressure (2% oxygen or treatment with CoCl₂ as compared to 20% oxygen pressure). These data have been discussed in the Results section at pag. 4.

From this analysis, and considering the model in Fig. 1h, it is unclear what is driving the increased expression of miR-30d under hypoxic conditions.

Data obtained from the experiments suggested by this reviewer and described here below, together with those suggested by Reviewer #3, demonstrate that expression of miR-30d increases under hypoxic conditions as a consequence of increased binding of HIF1 α and mut-p53 to *MIR30D* promoter. Specifically, we observed that under hypoxic conditions, HIF1 α becomes stabilized (Fig. 1e), and the binding of mut-p53 to HIF1 α increases of more that twofold, as shown by PLA assays (Fig. 1f). This likely causes the observed increase of mut-p53 recruitment at *MIR-30D* promoter, that is indeed HIF-dependent (Fig. 1g), as well as mut-p53-dependent deposition of the chromatin activation mark H3K9Ac (Fig. 1i), thereby inducing miR-30d expression.

Is it because HIF-1a binds better to the miR-30d chromatin under hypoxia?

This should be tested by performing a HIF1a ChIP under conditions similar to those used in Fig. 1g. As suggested, we performed ChIP for HIF1 α under conditions similar to those used in Fig. 1g. The results of these experiments, shown in new Fig. 1h, demonstrated an increased binding of HIF1 α to the *MIR30D* promoter region under hypoxia.

Moreover, the model in Fig. 1h predicts that more mut-p53 should be bound to the miR-30d locus under hypoxia than in normoxia. This should be tested by comparing the p53 ChIP as in Fig. 1g between 20% and 2%.

We performed the experiment as suggested. As shown in new Fig. 1g, binding of mut-p53 to *MIR30D* promoter was increased under hypoxic conditions in a HIF1 α -dependent manner.

3. Fig. 5f: the effect of miR-30d on VEGFA secretion by mut-p53-depleted cells is rather marginal, suggesting that most (90%?) of the effect of mut-p53 is not due to miR-30d. The authors should quantify the band intensities. As a matter of fact, Fig. 5h also shows only a very partial phenotypic rescue by miR-30d, which may be consistent with the limited effect of miR-30d on VEGFA secretion. Together, this suggests that much of the effect of mut-p53 on VEGFA secretion and activity is NOT via miR-30d, unlike what one may conclude from reading the text. The authors should discuss this issue.

As this reviewer correctly points out, the impact of mut-p53 on VEGFA activity involves additive effects. Indeed, VEGFA has been previously shown to be induced by mut-p53 at the transcriptional level (Pruszko et al., 2017). We have confirmed this result in MDA-MB-231 cells, and have added it to the manuscript (new Suppl. Fig.7d). In addition, the effect of mut-p53 and miR-30d on VEGFA secretion has been quantified (Fig. 5f). Taken together, these data can explain the partial phenotypic rescue of VEGFA secretion by miR-30d (which does not affect directly its expression) upon knockdown of mut-p53 (which induces both expression and secretion of VEGFA).

Minor comments:

1. Fig. 1f: the IgG control lane in the 20% panel seems to have a bubble, making the expected negative result less convincing. It will be helpful if the authors can provide a cleaner figure, showing clearly that there is no HIF1a band in the IgG control.

We have substituted Fig. 1f with cleaner co-IP experiments performed in both MDA-MB-231 and MDA-MB-268 cells with clear negative controls (new Suppl. Fig. 1j, k), moreover we have performed proximity ligation assays (PLA) to confirm that the interaction occurs already under normoxic conditions and is increased under hypoxic conditions (shown in Fig. 1f).

2. Fig. 3a: The PDI staining in the mut-p53 R280K cells look very different from that in the miR-30d cells. The authors need to comment on this, or provide a more representative image if these differences are not representative.

In general, the milder phenotypes observed upon expressing mut-p53 as compared to those observed upon ectopic expression of miR-30d can be explained by the fact that mut-p53 induces expression of smaller amounts of endogenous miR-30d as compared to those obtained upon ectopic overexpression of miR-30d. This said, we introduced a new picture in Fig. 3a that is more representative of the results observed.

3. Supplemental Fig. 5c suggests that miR-30d overexpression causes downregulation of mut-p53, but this is not seen in Supplemental Fig. 5e. Please explain.

It has been previously shown that miR-30d can specifically target the 3'UTR of p53 mRNA, leading to reduced expression of wild-type p53 (Kumar et al., *Oncogene*, 2011), and we have also observed this (suppl. Fig. 5a). In our hands however, the effect of miR-30d on mut-p53 expression was very mild, most likely due to the well-known high stability of mut-p53. For consistency, we have substituted the WB in new Supplementary Fig. 6c with a more representative picture.

4. Line 181: 3e should be 2e.

5. Line 333; 5a should be 5b.

6. Line 369: 'ad' should be 'and'

Reviewer #2 (Remarks to the Author):

NCOMMS-19-40638

Mutant p53 induces Golgi tubulo-vesiculation driving a prometastatic secretome
Valeria Capaci, Lorenzo Bascetta, Marco Fantuz, Galina Beznoussenko, Roberta Sommaggio, Andrea Bisso, Alexandre Mironov, Valeria Cancila, Jacek Wisniewski, Luisa Ulloa Severino, Denis Scaini, Fleur Bossi, Jodi Lees, Noa Alon, Ledia Brunga, David Malkin, silvano Piazza, Licio Collavin, Antonio Rosato, Silvio Bicciato, CLAUDIO TRIPODO, Fiamma Mantovani, and Giannino Del Sal.

While p53 is one of the most critical tumor suppressor, mutant p53 serves as an oncoprotein in Li-Fraumeni cancer predisposition syndrome. In this study, Dr. Del Sal and colleagues determined the role of mut-p53 in tissue culture cells, animal models and human tissues. They found that p53 mutants interact with the hypoxia responsive factor HIF1a and induce the expression of miR-30d, which reduces the DGKZ and VPS26B protein levels and causes tubulo-vesiculation of the Golgi apparatus. Subsequently, this enhances vesicular trafficking and secretion and promotes secretion of growth factors such as VEGF and ECM proteins. Thus, the authors have discovered a novel mut-p53/HIF1a/miR-30d axis that impacts synthesis, trafficking, and secretion of a number of proteins essential for tumor growth and metastatic colonization.

Overall the topic is novel and significant, and a large amount of data was provided, covering a broad range of aspects of cancer cell biology. However, some of the conclusions are based on weak results or speculations. There is also some critical information missing in the manuscript.

Specific points:

1. The interaction of mut-p53 and HIF1 is new, but some critical questions are not addressed.
1) Figure 1e, can HIF over expression rescue the sip53 effect?

We agree with this comment, however instead of ectopic over-expression of HIF1 α we have induced physiological HIF1 α stabilization by growing the cells at 2% oxygen pressure. In Fig. 1e we show that stabilization of HIF1 α by hypoxia is not sufficient to fully rescue the expression of miR-30d caused by mut-p53 knockdown.

Consistently with this evidence, in our ChIP experiments we observed that the presence of mut-p53 fosters the recruitment of HIF1 α to *MIR30D* promoter (new Figure 1h). This result could be explained by the notion that HIF1 α stability is modulated by mut-p53 (Montagner et al., 2012), and indeed in Fig. 1e we observed that mut-p53 knockdown leads to reduced stabilization of HIF-1 α by hypoxic conditions. In agreement with our results, Madan et al. recently demonstrated that wt-p53 with altered conformation and mut-p53 proteins bind and chaperone HIF1 α , stabilizing its binding to hypoxia responsive elements, and increase HIF transcriptional ability (Madan et al., 2019).

2) Figure 1f, HIF1 activity is known to be higher at low oxygen level, if the authors' hypothesis is that mut-p53 acts as a partner of HIF1 α , why the interaction is lower at low oxygen?

We agree, and indeed the results shown in the original Fig. 1f (panels showing co-IP under normoxic or hypoxic conditions) cannot be directly compared, since these were derived from independent experiments. To compare the interaction between endogenous mut-p53 and HIF1 α proteins under normoxic and hypoxic conditions, we performed proximity ligation assays (PLA) upon growing cells under normoxic (20% pO₂) or hypoxic (2% pO₂) conditions, as well as upon hypoxia-mimetic treatment with CoCl₂. The results of these new experiments, shown in new Figure 1f, highlighted that the binding of mut-p53 with HIF1 α increases of more than two-fold under low oxygen pressure (2% oxygen or treatment with CoCl₂ as compared to 20% oxygen pressure). These data have been discussed in the Results section at pag. 4.

2. The manuscript did not provide an adequate explanation of how the Golgi becomes vesiculated.

1) Fig. 3a, the shp53+p53 R280K cells have a different phenotype from miR-30d cells for beta-COP and GM130. What's the explanation?

In general, the milder phenotypes observed upon expressing mut-p53 as compared to those observed upon ectopic expression of miR-30d can be explained by the fact that mut-p53 induces expression of smaller amounts of endogenous miR-30d as compared to those obtained upon ectopic overexpression of miR-30d. This said, we introduced a new picture in Fig. 3a that is more representative of the average results observed.

2) The authors identified two proteins, DGKZ and VPS26B, that may affect Golgi morphology and protein trafficking, but it is not clear how. DGKZ is an enzyme that affects DAG to PA conversion, there is no mechanism for how it affects Golgi structure and function. VPS26B is involved in retrograde trafficking, no mechanism was provided whether it also affects anterograde trafficking and secretion.

miRNAs are a class of molecules able to finely modulate cellular processes at different levels, acting on multiple targets along the same pathway (Pasquinelli, 2012). Here we identified two direct targets of miR-30d, DGKZ and VPS26B, whose modulation affects Golgi structure. As

specified in the text, both proteins have been previously linked to modulation of GA structure and secretion (Aranda et al., 2018; Chia et al., 2012; Cui et al., 2019; Gharbi et al., 2013).

The DGKZ kinase is known to reduce the membrane levels of diacylglycerol (DAG), by phosphorylating DAG to phosphatidic acid (PA) (Asp et al., 2009; Yang et al., 2008). We demonstrated that via inhibiting DGKZ, miR-30d leads to accumulation of DAG in Golgi membranes (suppl. fig. 6g). Increased levels of DAG have been shown to affect Golgi structure and functions by two major mechanisms, i.e. modulating membrane curvature due to its conical shape (Bard and Malhotra, 2006) and by activation of Golgi resident PKD kinase (Bankaitis et al., 2012; Roy et al., 2017), thereby facilitating vesicular transport and secretion. In particular, variations in the local concentration of DAG have been previously shown to regulate alterations of Golgi structure known as tubulo-vesiculation (Asp et al., 2009) and to promote trafficking from trans-Golgi to plasma membrane (Baron and Malhotra, 2002). Concerning PKD1 activation at Golgi membranes, we demonstrated that by targeting DGKZ, miR-30d leads to local activation of PKD, as shown by increased phospho-PKD in Golgi apparatus (fig. 4g). All these data provide mechanistic explanation of the role of DGKZ in affecting Golgi structure and increasing secretion.

The second target, VPS26B, is a component of the core retromer complex that regulates endosomal recycling and retrograde transport from endosomes to the trans-Golgi network (Gallon and Cullen, 2015; Kerr et al., 2005). We demonstrated that miR-30d decreased VPS26B expression (fig. 4b, Supplementary fig. 6a,c,e) and consistently observed that VPS26B inhibition altered Golgi structure and induced secretion (fig. 4e-f). Defects in retromer activity are known to attenuate retrograde transport from endosomes to Trans-Golgi network (TGN): this alteration results in inappropriate endosomal sorting to intracellular destinations (Aranda et al., 2018; Burd and Cullen, 2014), impacting on its recycling activities with consequent increased secretion. This has been previously documented for cathepsin-D precursor (Cui et al., 2019), and A β in Alzheimer's Disease (Sullivan et al., 2011). In addition, because of impaired traffic to TGN from endosomes, retromer dysfunction induces structural and functional alteration in Golgi (Aranda et al., 2018). All this evidence might explain the role of miR-30d in affecting trafficking and GA morphology via targeting VPS26B, and we have now explained these concepts more clearly in the Discussion section. It would be certainly interesting to deepen the analysis of VPS26B-retromer complex activity in secretion, however this analysis goes well beyond the scope of this work, and will be the subject of future studies.

3) The authors should at least test a few key Golgi structural and trafficking proteins (e.g. Sec24) by Western blot in cells of the 4 different treatments as in Figure 3A and see which protein is affected. Results from tissue culture cells should be confirmed in mouse and human tissues.

We performed the requested experiments, employing MCF10A cells as in previous Figure 3a. We monitored the expression of components of ER, GA, COPI-II transport vesicles and microtubules by WB upon silencing endogenous wild-type p53 and ectopic overexpression of either mut-p53 R280K or miR-30d. As shown in new Suppl. Fig. 5a, silencing of wt-p53 did not affect the level of any of the proteins investigated, while protein levels of PDIA5, Sec24A and GM130 (protein markers for ER, COPI, and Golgi compartment respectively) were increased upon overexpression of either mut-p53 R280K or miR-30d, while the levels of tubulin and β -COP were not affected in this setting. These results have been discussed in the Results section at pag 7.

We have also confirmed results from tissue culture cells in mouse and human tissues. We performed immunoblot analysis of primary tumors from engrafted mice either with control vector or dy-30d construct, which highlighted a reduced expression of PDI, Sec24A, β -COP and GM130 upon miR-30d inhibition. These results have been included in new Suppl. Fig. 8a. We also performed immunohistochemical analysis of primary tumors, which highlighted that tumor cells of control injected mice displayed enlarged Golgi as judged by GM130 staining, while miR-30d inhibition rescued normal perinuclear Golgi staining. These results have been included in new Fig. 6b.

Finally, we performed immunohistochemical analysis of human breast cancer samples. Tumors bearing mut-p53 also displayed increased Golgi area, as judged by GM130 immunostaining, suggesting an enlarged organelle; moreover, immunostaining showed a consistent increase of Sec24A and β -COP in samples bearing mut-p53, suggesting enhancement of the secretory trafficking. These results have been included in new Fig 7f-g-h.

3. It is not clear how Golgi vesiculation enhances protein trafficking and secretion.

The mechanism of transport through the Golgi complex is still a matter of discussion among experts in the field, and it has been proposed that different mechanisms of intra-Golgi trafficking may co-exist.

In particular, it has been previously shown that soluble secretory proteins diffuse across the Golgi stack in the *cis-trans* direction through inter-cisternal tubules. These are narrow tunnel-like connections between individual cisternae, characterized by dynamic and transient nature (Trucco et al., 2004). Soluble proteins, moving through these structures, traverse the Golgi much quicker than non-diffusible cargoes (large protein complexes and trans-membrane proteins) that cannot enter these tubules and traverse the stack slowly moving by compartment progression–maturation (Beznoussenko et al., 2014).

Interestingly, structural adaptations of trafficking hubs including tubulo-vesiculation of the GA and expansion of the Golgi network that allow faster cargo diffusion have been associated both in physiological and pathological contexts (including cancer) with increased flux of secreted proteins (Halberg et al., 2016; Howley et al., 2018; Plate and Wiseman, 2017; Stalder and Gershlick, 2020; Taniguchi and Yoshida, 2017; Trucco et al., 2004).

Our results show that mut-p53/miR-30d overexpressing cells display Golgi tubulo-vesiculation, suggesting that this axis induces adaptation to allow quicker diffusion and transport of soluble proteins in tumor cells, which may have a profound impact in tumor tissues.

1) It is not clear how Golgi vesiculation enhances protein trafficking or secretion. The authors speculated that microtubule destabilization or degradation may cause Golgi fragmentation, but this is inconsistent with the fact that destabilization of microtubules slows down protein trafficking, instead of accelerating it.

We agree and we clarified this concept in the text. In fact, we did not speculate that microtubule destabilization or degradation may cause Golgi fragmentation. Instead, we observed that the mut-p53/miR-30d axis induces moderate stabilization (not destabilization) of microtubules, and we have now included a new Figure 3a that shows this effect more clearly. As pointed out by the reviewer, this is consistent with increased Golgi trafficking, and indeed we observed that treatment of cells with the microtubule stabilizing agent taxol increases Golgi vesiculation and protein secretion (Suppl. Fig. 6k, Suppl. Fig. 6l). Moreover, we observed that taxol could partly rescue the effect of miR-30d inhibitor on ssGFP secretion. This part has been clarified in the Results section at pag. 10, and new data have been added to Suppl. Fig. 6.

2) Changes in trafficking may not always impact secretion depending on the time when they are determined. It is not described how long the cells were incubated before ssGFP was collected in the medium, but it was shown that at this time most ssGFP was still inside the cells. Did the authors include cycloheximide in this experiment? Since most protein trafficking from the ER to the plasma membrane takes about 2 hours, where is ssGFP stuck in cells? How long were the cell incubated when the secretome was analyzed?

We agree, and indeed to compare secretion under different experimental conditions, we chose a time frame of 2 hrs, i.e. cells expressing ssGFP were incubated with fresh medium that was collected after 2 hours, and the ssGFP protein secreted in the medium during this time frame was then analyzed, compared to the amount of intracellular GFP. This has been now made clear in the

Fig. legends (Fig. 2e, Fig. 7d, Suppl. Fig. 4b, and Suppl. Fig. 6j-l). Since secretion experiments were not performed in the presence of cycloheximide, intracellular ssGFP was continuously produced during the course of the experiment, and therefore it is found within cells even after 2 hours. Importantly, ssGFP is not found stuck in a particular localization within the cells, rather, upon translation, it enters the ER compartment and localizes along the whole secretory route (ER, COP vesicles and GA) prior to being secreted. We have included a new Suppl. Fig. 4a to make this clear.

3) In general, secretory proteins are either degraded by ERAD if misfolded or secreted. Their trafficking speed may be modulated by the trafficking machineries, but they are normally not held in the Golgi.

We agree, and indeed we propose that trafficking speed is increased by adaptation of the trafficking machinery, i.e. Golgi tubulo-vesiculation. As discussed in the above point, we do not observe that proteins are held in the Golgi, rather they are distributed along the whole secretory route (ER, COP vesicles and GA) prior to being secreted.

The comparison of secretome with the transcriptome from a different study in Figure 2h is not valid as these two studies may have different experimental conditions or settings.

In our work we have compared transcriptome, proteome, and secretome data, all generated in our lab, from aliquots of the same cell line (MDA-MB-231) and with the same siRNA reagents, with identical experimental conditions and settings (Walerych et al., 2016). Importantly, these data were instrumental for interpreting our secretome data. This generated the hypothesis that mut-p53 might impact the cell secretory machinery and enhance secretion, which then represented a major focus of our manuscript, supported by experimental evidence.

4) Is accelerated trafficking and enhanced secretion selective for some proteins but not others? This should be confirmed with a few target proteins.

This is an interesting question. In fact, work by (Beznoussenko et al., 2014; Trucco et al., 2004), suggests that Golgi tubulo-vesiculation may favor transport of soluble proteins by accelerating diffusion through continuities across cisternae, while it may not equally favor secretion of membrane proteins or large protein complexes. In fact, membrane proteins were not included in our secretome analysis. We have now discussed this issue in the text (Results section, pag. 8).

However, to test the possibility that trafficking of transmembrane proteins may be reduced by mut-p53 or miR-30d we used a reporter for canonical secretion of transmembrane proteins, i.e. the ssHRP-TM reporter (Li et al., 2010), generated fusing the signal peptide of human CD2 protein at the N-ter of the horseradish peroxidase protein, followed by a transmembrane domain and a perimembrane region. We chose this strategy to avoid individual specificity by testing a bunch of secreted proteins. Our results, shown in the figure below, indicate that both mut-p53 and miR-30d reduce the trafficking of ssHRP-TM. Discussing these results in the text would however require validation and generalization of this result, which implies performing a large number of experiments, which have been hampered by overall experimental restrictions that we are suffering during this period, and goes beyond the scope of this work.

Fig.1 miR-30d and mut-p53 inhibit secretion of ssHRP-TM. H1299 cells were transfected with a plasmid encoding the horseradish peroxidase protein fused to a secretion signal sequence and followed by a transmembrane domain (ssHRP-TM), the plasmid expresses also a green fluorescent protein under the same promoter. 24 h upon miR-30d and mut-p53 overexpression, surface HRP enzymatic activity has been evaluated by the Ultra TMB-ELISA kit (Life Technologies). GFP, Actin and p53 were blotted as control.

5) Related to above, it is necessary to figure out whether the reduced secretion of specific proteins is because of reduced synthesis or trafficking.

In our work we did not focus on secretion of specific proteins, rather we highlighted a general mechanism leading to global increase of protein secretion. However, as shown in Suppl. Fig. 3c-d-e-f-g, analysis by metabolic labeling did not highlight major differences in the synthesis of intracellular proteins that might explain changes in secretion.

6) Figure 3d, ManII is a Golgi marker and so can only be used to measure ER-to-Golgi trafficking, not intra-Golgi trafficking related to Golgi vesiculation. In this figure, the Golgi does not seem to be vesiculated in the miR-30d cells. There is also no quantitation of the movie.

MannII is a medial-trans Golgi protein marker (Bossard et al., 2007; Rabouille et al., 1995), hence it can be used also to measure intra-Golgi trafficking.

Since live imaging is not confocal, the resolution is reduced as compared to Fig.3, Suppl. Fig. 5, therefore vesiculation is less evident. Quantitation of the MannII movie has been added to new Fig3e.

7) Movie 2 is not convincing. Since E-Cad is N-glycosylated, the authors could use EndoH treatment and bandshift to indicate the trafficking speed.

This reviewer correctly points out that the effect of miR-30d on E-Cad trafficking is complex. The movie highlights that the entire trafficking from ER to Golgi exit is accelerated in cells overexpressing miR-30d as compared to control. However, this process includes three different steps (ER-Golgi transport, intra-Golgi transport and Golgi exit). The trafficking of E-cad from ER to Golgi compartment is accelerated of about 3-fold in cells overexpressing miR-30d as compared to control (20 min upon synchronization vs 60 min in control cells), and a faster Golgi exit of E-Cad is also observed (80 min vs 120 min). However, the intra-Golgi time of transit is of 60 min in both conditions, indicating that the rate of intra-Golgi trafficking could be unaffected. This is consistent with the notion that Golgi tubulation accelerates intra-cisternae diffusion of soluble proteins, while the transport of membrane proteins may be instead delayed (see above, point 4). Regrettably, due to experimental restriction associated to COVID-19 lockdown, it has not been possible for us to perform the experiments suggested by the reviewer nor to perform a new RUSH assay to extend the analysis of the trafficking rate till the plasma membrane. Therefore, since the conclusions of the work are supported by many other independent observations, we decided to remove Movie 2 and its description from the results.

8) Figure 4g, the pPKD pattern is odd, its localization to the Golgi is not convincing.

There are several published images of phospho-PKD localization pattern, e.g. (Eisler et al., 2018; Mazzeo et al., 2016) that show a phospho-PKD Golgi localization pattern consistent with the staining that we observed in MDA-MB-231 cells.

4. Some critical controls or details are missing.

1) miR-30d was previously identified as a negative regulator of p53 (Kumar et al., *Oncogene*, 2011). This was not discussed in this study.

It has been previously shown that miR-30d can specifically target the 3'UTR of p53 mRNA, leading to reduced expression of wild-type p53 (Kumar et al., 2011), and we have also confirmed this (suppl. Fig. 5a). In our hands however, the effect of miR-30d on mut-p53 expression was very mild, most likely due to the well-known high stability of mut-p53. For consistency, we have substituted the WB of new Suppl. Fig. 6c with a more representative picture.

2) On the other hand, several known microRNAs regulated by p53 such as miR-34 were not included in the screening.

As described in the text, only oncogenic miRNAs were included in the screening. Among them, as a control we included the mut-p53 target mir-128 (Donzelli et al., 2012). In contrast, miR-34 is a tumor suppressor miRNA that is induced by wild-type p53, and therefore it was not included in the screening.

3) In addition to the p53 mutants, WT p53 should be tested in parallel in some of the critical experiments.

The effects of wild-type p53 have been tested in all critical experiments by silencing wild-type p53 by RNA interference.

First, we tested the effect of wt-p53 knockdown on miR-30d expression in HBL100 and MCF-7 cancer cells, (Suppl Fig 1c) as well as in MCF10A normal-like breast epithelial cells (fig. 1c, Suppl Fig 1c): in these experiments, endogenous wt-p53 did not appear to regulate the expression of miR-30d.

We tested the effect of wt-p53 knockdown on protein secretion by analyzing ssGFP (fig. 2a) and by metabolic labeling (suppl. Fig. 3f): in these experiments, depletion of endogenous wt-p53 had no significant effect on total protein secretion.

Finally, we analyzed the effect of depleting endogenous wt-p53 on GA structure in: fig. 3 a, b, d: in these experiments, wt-p53 knockdown did not cause significant alterations of GA structure.

Notably, all the above experiments were also performed upon treatment of mut-p53 expressing cells with the compound APR-246/PRIMA-1MET, which is able to restore wt-p53 function (ref). In these experiments, effects of PRIMA were equivalent to knockdown of mut-p53, suggesting no effect of wild-type p53 on the phenotypes analyzed (suppl fig. 4f, fig. 3c, suppl. Fig 5f).

4) The authors collected conditioned medium from MDA-MB-231 and H1299 cells, but did not mention how long the cells were incubated before the conditioned medium was collected. In addition, MDA-MB-231 cells were cultured in DMEM while H1299 cells were cultured in RPMI1640; the two different media have different compositions.

We apologize for having missed this information, that has now been included in the legend of Fig. 5d and in the Methods section "Preparation of conditioned medium". Specifically, in the experiments shown in Fig. 5d-e, both MDA-MB-231 donor cells and H1299 donor cells were cultured in the same conditions and with the same growth medium, i.e. DMEM w/o FBS for 48h to produce conditioned medium.

5) Molecular weight markers are missing for some gels, scale bars are missing in some micrographs.

These have been added to all gels and micrographs.

6) For the proteomic and bioinformatic studies it would be helpful to indicate what proteins/genes are included in which categories and how much they have changed under different conditions. Some of the tables include the protein/gene names but did not show which categories they belong to. The requested data have been included in Supplementary tables 1-2.

Reviewer #3 (Remarks to the Author):

This is an interesting manuscript that documents a novel gain of function activity of mutant p53 proteins in breast cancer. The authors intriguingly show that the transcriptional axis mutp53/HIF1 α promotes aberrant activation of miR-30d that provokes the release of a pro-metastatic secretome. The listed below comments need to be fully addressed to render the manuscript more logically organized and more balanced between in vitro and in vivo reported findings.

Specific comments:

1. Most of the reported in vivo experiments refer to the mutp53-miR30d axis lacking any evidence that includes HIF1 α . This is not consistent with the transcriptional regulatory role that mutp53 (HIF1 α exerts on miR-30d expression; thereby providing a set of data lacking the main transcriptional and functional player such as HIF1 α).

The experiments performed in vivo in mice analyzed the effects of miR-30d, rather than addressing the regulation of miR-30d expression by the mut-p53/HIF1 α axis, therefore neither HIF1 α nor mut-p53 were included as variables in this experimental setting (Fig. 6).

However, to answer the reviewer's request, we have now introduced a large set of new experimental data to address more deeply the role of HIF-1 α in mut-p53/miR-30d phenotypes (see also points 5 and 8). In particular, we have analyzed:

- Effect of HIF1 α on secretion: consistently with our observation that HIF1 α is a major inducer of miR-30d expression, we demonstrated that knockdown of HIF-1 α dampened ssGFP secretion, whereas hypoxic conditions, that stabilize HIF-1 α , promoted secretion in a miR-30d dependent manner. These results have been included in new Figure 2g.
- Effect of HIF1 α on Golgi structure: we demonstrated that hypoxic conditions induced a vesiculated GA morphology in MCF-10A cells, that was reverted upon inhibition of miR-30d, indicating that HIF1 α alters GA structure and function via inducing miR-30d. These data have been included in new Suppl. Fig. 5g.
- Role of HIF1 α in the effect of cancer cell secretome on stromal cells: we demonstrated that conditioned medium obtained from MDA-MB-231 cells grown under hypoxic conditions enhanced fibroblasts activation, and this effect was reduced upon miR-30d inhibition in donor cells, suggesting that HIF-1 α induces a fibroblast-activating secretome via induction of miR-30d expression. These results have been included in new Suppl. Fig. 7h.

2. Do breast cancer patients carrying TP53 mutations in Metabric and TCGA or other datasets exhibit higher levels of miR-30d? What about the reported and investigated miR-30d targets? Is their expression anti-correlated to miR-30d?

Analysis of tumor-derived gene expression datasets highlighted a correlation of high miR-30d expression with TP53 mutation (Metabric) and with high HIF1 α levels (TCGA BRCA). This evidence has been included in new Suppl. Fig. 1m, n and discussed in the results section at pag. 4.

Regarding correlation of miR-30d with the expression of its targets DGKZ and VPS26B, we found that expression of VPS26B mRNA is significantly anti-correlated with miR-30d levels, both in Metabric and TCGA BRCA datasets. Conversely, we did not find any significant correlation or

anti-correlation between miR-30d levels and DGKZ mRNA expression in these datasets. However, miRNAs frequently control their target mRNAs by inhibiting translation rather than inducing mRNA degradation. Therefore, analyzing anticorrelation of miR-30d expression with that of its targets should also include the analysis of target protein levels. Unfortunately, we could not perform this analysis for DGKZ due to very limited information available in protein expression databases. Instead, we analyzed primary fibroblasts derived from Li-Fraumeni patients carrying germline missense *TP53* mutations as compared to healthy controls: in these samples we observed association of mut-p53 and HIF1 α expression with increased miR-30d levels, and reduced levels of VPS26B and DGKZ. These data have been included in Fig. 7.

3. *TP53* sequencing and LOH analyses of the 12 analyzed human breast cancers is required to confirm that they are representative of specific mutant p53 gain of function tumors. P53 immunostaining isn't sufficient for making the conclusion proposed by the authors.

To confirm the presence of specific missense *TP53* mutations in the tumors analyzed, we performed competitive Allele-Specific TaqMan PCR. All p53-high cases, bearing high levels of p53 expression, displayed missense *TP53* mutations, including well-known hot-spot mutations. In contrast, none of p53-low cases, displaying low or null p53 expression, contained *TP53* mutations as highlighted by the assay. These data have been summarized in Supplementary table 3.

4. It appears rather unclear why adult fibroblasts from Li-Fraumeni represent a unique genetic model to study the impact of mutp53/miR-30d axis on breast tumorigenesis. If any, it's difficult to believe that fibroblasts can recapitulate the complexity of established breast cancer tumors. One intriguing possibility could be that of IHC of breast cancer tissues from LFS patients.

We agree with this comment. We meant that primary cells from non-tumor tissues of Li-Fraumeni patients represent a unique genetic model to study the impact of germline *TP53* mutations on miR-30d and its target in human non-tumor tissues, and we have now clarified it in the text. These results are important since they show that the mut-p53/HIF1 α /miR-30d axis may be active also in these patients in a preneoplastic condition and may thereby impact on the onset of phenotypes associated with this syndrome.

Analyzing tumor tissues from LFS patients could have added information to this study, however the patients analyzed did not develop breast cancers. Furthermore, due to extreme rarity of Li-Fraumeni syndrome, it was not possible to obtain breast tumor samples from other Li-Fraumeni patients. We have therefore analyzed breast cancer samples bearing sporadic *TP53* mutations, as described in Figure 7.

5. Does HIF1 α depletion affect the mutp53-miR30d dependent secretome? While the authors initially showed that miR30d expression is controlled by the transcriptional crosstalk mutp53/HIF1 α , most of the in vitro and vivo biological experiments have been performed lacking any evidence on the role of HIF1 α in mutp53-miR30d axis.

Consistently with our observation that HIF1 α is a major regulator of miR-30d expression, we demonstrated that hypoxic conditions induced a vesiculated GA morphology in MCF-10A cells, that was reverted upon inhibition of miR-30d. These results demonstrate that HIF-1 α alters GA structure and function via inducing miR-30d have been included in Suppl. Fig. 5g.

To address the reviewer's question as to whether HIF-1 α depletion affects mut-p53/miR-30d dependent secretion, we performed knockdown of HIF-1 α in MDA-MB-231 cells and observed reduced ssGFP secretion, whereas in these cells hypoxic conditions promoted secretion in a miR-30d dependent manner. These results have been included in new Figure 2g.

Moreover, to demonstrate that alteration of HIF-1 α levels affects the mut-p53/miR-30d secretome, we performed experiments with conditioned medium obtained from MDA-MB-231 cells grown either under normoxic or hypoxic conditions. We demonstrated that growth of MDA-MB-231 donor cells under hypoxic conditions (stabilizing HIF-1 α levels) increased the ability of their

secretome (CM) to enhance fibroblast activation, and this effect was reduced upon miR-30d inhibition in donor cells. These data suggest that HIF-1 α promotes the release of a tumor supportive secretome by cancer cells under hypoxia via induction of miR-30d expression. These results have been included in Suppl. Fig. 7h.

6. How the authors explain that mutap53/HIF1 α bind miR30d regulatory regions in both normoxic and hypoxic conditions? The described transcriptional complex is active in both conditions? What about the analysis of markers of chromatin modifications or the recruitment of acetylase or deacetylase in both conditions to address this issue.

To understand the contribution of the transcriptional complex formed by mut-p53 and HIF1 α to the activation of *MIR30D* promoter in normoxic and hypoxic conditions, we decided to characterize the formation of a complex between the two factors and their interaction with chromatin, as well as their impact on promoter activity.

To this aim we performed proximity ligation assays (PLA) and chromatin immunoprecipitation experiments for mut-p53, HIF1 α and the chromatin activation mark H3K9Ac under either normoxic (20% pO₂) or hypoxic (2% pO₂) conditions. The results have been included in new Figure 1 and discussed in the Results section at pag. 4.

Specifically, PLA and co-IP experiments highlighted that interaction between mut-p53 and HIF1 α can be detected already under normoxic conditions (Fig. 1f, Suppl. fig. 1k). Consistently, ChIP experiments highlighted that specific binding of both mut-p53 and HIF1 α to *MIR-30D* promoter occurs at low levels already under normoxic conditions (shown in new Fig. 1g, h). As shown in Figure 1f, in these conditions mut-p53 is required for promoter activation as judged by its ability to promote deposition of the chromatin activation mark H3K9Ac.

Under hypoxic conditions, HIF1 α becomes stabilized (Fig. 1e), and the binding of mut-p53 to HIF1 α increases of more that twofold, as shown by PLA assays (Fig. 1f). This likely causes the observed increase of mut-p53 recruitment at *MIR-30D* promoter, that is indeed HIF-dependent (Fig. 1g), as well as mut-p53-dependent deposition of the chromatin activation mark H3K9Ac (Fig. 1i). Interestingly, HIF1 α stabilization by hypoxia (Fig. 1e) and consequent promoter binding (Fig. 1h) are further promoted by mut-p53.

Altogether, these results demonstrate that mut-p53 and HIF1 α form an active transcriptional complex on *MIR-30D* promoter, leading to miR-30d expression already in normoxic conditions, and to its further increase under hypoxic conditions.

7. Recruitment of HIF1A on miR-30d promoter needs to be analyzed as well, to conclude that mut-p53 acts as a partner of HIF1 α , leading to transcriptional upregulation of miR-30d expression.

As described in point 6, this has been analyzed leading to the conclusions reported above.

8. Effect on secretion after HIF1A stabilization in presence/absence of miR-30d inhibition needs to be analyzed to conclude that the mutp53/HIF1A complex impinges on secretion through miR-30d induction and activity.

As already specified at point 1 and 5, to verify that the mut-p53/HIF1 α complex impinges on secretion through miR-30d induction, we verified that stabilization of HIF1 α by hypoxic conditions promoted ss-GFP secretion in a miR-30d dependent manner. These results have been included in new Fig. 2g.

9. Golgi vesiculation has been mainly analyzed in non-transformed cells throughout the manuscript with the majority of results obtained in MCF10A cells and fibroblasts. The relevance of the identified network in cancer cells could be strengthened by the inclusion of additional analyses performed in cancer cells after modulation of the key players of the identified network.

As requested by the reviewer, we analyzed the effect of the mut-p53/miR-30d axis on GA structure in cancer-derived cell lines. In particular, we demonstrated that ectopic expression of mut-p53 variants R175H, R273H and R280K in p53-null H1299 lung cancer cells induced GA alteration that was abrogated by inhibiting endogenous miR-30d (Suppl. Fig. 5e).

Moreover, MDA-MB-231 and MDA-MB-231 and MDA-MB-231 cells, harboring different endogenous mut-p53 variants, displayed a vesiculated GA morphology that was normalized upon knock-down of mut-p53 or treatment with the mut-p53 inactivating agent PRIMA-1MET (Lambert et al., 2009), and reverted by introduction of miR-30d (Fig. 3c and Suppl. Fig. 5f).

Minor comments:

1. Figure 2, was GSEA performed only considering the protein secretion pathway? Information provided in the results/legend for GSEA analysis is incomplete (q-val? Enrichment?).

The requested values have been added in figure 2b.

2. Does miR-30d impact on proliferation?

What about mutp53? If this is the case, secretion should be analyzed also after additional cell cycle arresting stimuli.

Effects of mut-p53 on cell proliferation are well known (Walerych et al., 2015); in addition, we have observed a moderate effect of miR-30d on cell proliferation. However, ssGFP and metabolic labeling experiments were performed within a time frame of 2 h, and therefore the results are independent on variations of cell proliferation. Moreover, analysis of the results was relative to intracellular protein content, thereby normalizing for cell number. Instead, preparation of conditioned medium that required a time frame of 48 h was performed upon serum starvation as cell-cycle arresting stimulus.

3. Fig2g, the description is not clear, please include in the matrix the control samples siSCR and miR-CTR

In this figure, samples sip53/CTRL and sip53 + miR-30d/CTRL refer to changes relative to CTRL condition. This has been better clarified in the figure.

4. Fig2. Venn diagram of proteins controlled by mp53 and by miR-30 would help in understanding results

The proteomic analysis has been performed by LC MS/MS technology on CMs collected from control and mut-p53-KD MDA-MB-231 cells, and from same cells overexpressing miR-30d. A Venn diagram would not be informative as it would display that the subset of proteins whose secretion is restored by reintroduction of miR-30d is included within the set of proteins regulated by mut-p53. To add the requested information, we have included the list of differentially secreted proteins, with relative changes in all conditions, in supplementary table 2.

References

Aranda, J.F., Rathjen, S., Johannes, L., and Fernández-Hernando, C. (2018). MiR-199a-5p attenuates retrograde transport and protects against toxin-induced inhibition of protein biosynthesis. *Mol. Cell. Biol.* MCB.00548-17.

Asp, L., Kartberg, F., Fernandez-Rodriguez, J., Smedh, M., Elsner, M., Laporte, F., Bárcena, M., Jansen, K.A., Valentijn, J.A., Koster, A.J., et al. (2009). Early stages of Golgi vesicle and tubule formation require diacylglycerol. *Mol. Biol. Cell* 20, 780–790.

Bankaitis, V.A., Garcia-Mata, R., and Mousley, C.J. (2012). Golgi membrane dynamics and lipid metabolism. *Curr. Biol.* 22, R414–R424.

- Bard, F., and Malhotra, V. (2006). The Formation of TGN-to-Plasma-Membrane Transport Carriers. *Annu. Rev. Cell Dev. Biol.* 22, 439–455.
- Baron, C.L., and Malhotra, V. (2002). Role of diacylglycerol in PKD recruitment to the TGN and protein transport to the plasma membrane. *Science* (80-). 295, 325–328.
- Bezoussenko, G. V., Parashuraman, S., Rizzo, R., Polishchuk, R., Martella, O., Giandomenico, D. Di, Fusella, A., Spaar, A., Sallese, M., Capestrano, M.G., et al. (2014). Transport of soluble proteins through the Golgi occurs by diffusion via continuities across cisternae. *Elife* 2014, 1–27.
- Bossard, C., Bresson, D., Polishchuk, R.S., and Malhotra, V. (2007). Dimeric PKD regulates membrane fission to form transport carriers at the TGN. *J. Cell Biol.* 179, 1123–1131.
- Burd, C., and Cullen, P.J. (2014). Retromer: A master conductor of endosome sorting. *Cold Spring Harb. Perspect. Biol.* 6, 1–14.
- Chia, J., Goh, G., Racine, V., Ng, S., Kumar, P., and Bard, F. (2012). RNAi screening reveals a large signaling network controlling the Golgi apparatus in human cells. *Mol. Syst. Biol.* 8, 1–33.
- Cui, Y., Carosi, J.M., Yang, Z., Ariotti, N., Kerr, M.C., Parton, R.G., Sargeant, T.J., and Teasdale, R.D. (2019). Retromer has a selective function in cargo sorting via endosome transport carriers. *J. Cell Biol.* 218, 615–631.
- Donzelli, S., Fontemaggi, G., Fazi, F., Di Agostino, S., Padula, F., Biagioni, F., Muti, P., Strano, S., and Blandino, G. (2012). MicroRNA-128-2 targets the transcriptional repressor E2F5 enhancing mutant p53 gain of function. *Cell Death Differ.* 19, 1038–1048.
- Eisler, S.A., Curado, F., Link, G., Schulz, S., Noack, M., Steinke, M., Olayioye, M.A., and Hausser, A. (2018). A Rho signaling network links microtubules to PKD controlled carrier transport to focal adhesions. *Elife* 7, 1–30.
- Gallon, M., and Cullen, P.J. (2015). Retromer and sorting nexins in endosomal sorting. *Biochem. Soc. Trans.* 43, 33–47.
- Gharbi, S.I., Avila-Flores, A., Soutar, D., Orive, A., Koretzky, G.A., Albar, J.P., and Merida, I. (2013). Transient PKC shuttling to the immunological synapse is governed by DGK and regulates L-selectin shedding. *J. Cell Sci.* 126, 2176–2186.
- Halberg, N., Sengelau, C.A., Navrazhina, K., Molina, H., Uryu, K., and Tavazoie, S.F. (2016). P1TPNC1 Recruits RAB1B to the Golgi Network to Drive Malignant Secretion. *Cancer Cell* 29, 339–353.
- Howley, B. V., Link, L.A., Grelet, S., El-Sabban, M., and Howe, P.H. (2018). A CREB3-regulated ER–Golgi trafficking signature promotes metastatic progression in breast cancer. *Oncogene* 37, 1308–1325.
- Kerr, M.C., Bennetts, J.S., Simpson, F., Thomas, E.C., Flegg, C., Gleeson, P.A., Wicking, C., and Teasdale, R.D. (2005). A novel mammalian retromer component, Vps26B. *Traffic* 6, 991–1001.
- Kumar, M., Lu, Z., Takwi, A.A.L., Chen, W., Callander, N.S., Ramos, K.S., Young, K.H., and Li, Y. (2011). Negative regulation of the tumor suppressor p53 gene by microRNAs. *Oncogene* 30, 843–853.
- Lambert, J.M.R., Segerba, D., Bergman, J., Gorzov, P., Vepintsev, D.B., So, M., Fersht, A.R., Hainaut, P., Wiman, K.G., and Bykov, V.J.N. (2009). PRIMA-1 Reactivates Mutant p53 by Covalent Binding to the Core Domain. 376–388.
- Li, J., Wang, Y., Chiu, S.L., and Cline, H.T. (2010). Membrane targeted horseradish peroxidase as a marker for correlative fluorescence and electron microscopy studies. *Front. Neural Circuits* 4, 1–10.
- Madan, E., Parker, T.M., Pelham, C.J., Palma, A.M., Canas-marques, R., Peixoto, L., Nagane, M., Chandaria, A., Tom, A.R., Henriques, V., et al. (2019). HIF-transcribed p53 chaperones HIF-1 . 1–23.
- Mazzeo, C., Calvo, V., Alonso, R., Mérida, I., and Izquierdo, M. (2016). Protein kinase D1/2 is involved in the maturation of multivesicular bodies and secretion of exosomes in T and B lymphocytes. *Cell Death Differ.* 23, 99–109.
- Montagner, M., Enzo, E., Forcato, M., Zanconato, F., Parenti, A., Rampazzo, E., Basso, G., Leo, G., Rosato, A., Biciato, S., et al. (2012). SHARP1 suppresses breast cancer metastasis by promoting degradation of hypoxia-inducible factors. *Nature* 487, 380–384.
- Pasquinelli, A.E. (2012). MicroRNAs and their targets: Recognition, regulation and an emerging reciprocal relationship. *Nat. Rev. Genet.* 13, 271–282.
- Plate, L., and Wiseman, R.L. (2017). Regulating Secretory Proteostasis through the Unfolded Protein Response: From Function to Therapy. *Trends Cell Biol.* 27, 722–737.
- Pruszkowski, M., Milano, E., Forcato, M., Donzelli, S., Ganci, F., Di Agostino, S., De Panfilis, S., Fazi, F., Bates, D.O., Biciato, S., et al. (2017). The mutant p53-ID4 complex controls VEGFA isoforms by recruiting lncRNA MALAT1.

EMBO Rep. *18*, 1331–1351.

Rabouille, C., Hui, N., Hunte, F., Kieckbusch, R., Berger, E.G., and Warren, G. (1995). Mapping the distribution of Golgi enzymes involved in the construction of complex oligosaccharides. *1627*, 1617–1627.

Roy, A., Ye, J., Deng, F., and Wang, Q.J. (2017). Protein kinase D signaling in cancer: A friend or foe? *Biochim. Biophys. Acta - Rev. Cancer 1868*, 283–294.

Stalder, D., and Gershlick, D.C. (2020). Direct trafficking pathways from the Golgi apparatus to the plasma membrane. *Semin. Cell Dev. Biol.* 0–1.

Sullivan, C.P., Jay, A.G., Stack, E.C., Pakaluk, M., Wadlinger, E., Fine, R.E., Wells, J.M., and Morin, P.J. (2011). Retromer disruption promotes amyloidogenic APP processing. *Neurobiol. Dis.* *43*, 338–345.

Taniguchi, M., and Yoshida, H. (2017). TFE3 , HSP47 , and CREB3 Pathways of the Mammalian Golgi Stress Response. *36*, 27–36.

Trucco, A., Polishchuk, R.S., Martella, O., Pentima, A. Di, Fusella, A., Di, D., Pietro, E.S., Beznoussenko, G. V, Polishchuk, E. V, Baldassarre, M., et al. (2004). Secretory traffic triggers the formation of tubular continuities across Golgi sub-compartments. *6*.

Walerych, D., Lisek, K., and Del Sal, G. (2015). Mutant p53: One, No One, and One Hundred Thousand. *Front. Oncol.* *5*.

Walerych, D., Lisek, K., Sommaggio, R., Piazza, S., Ciani, Y., Dalla, E., Rajkowska, K., Gaweda-walerych, K., Ingallina, E., Tonelli, C., et al. (2016). Proteasome machinery is instrumental in a common gain-of-function program of the p53 missense mutants in cancer. *18*.

Yang, J., Gad, H., Lee, S.Y., Mironov, A., Zhang, L., Beznoussenko, G. V, Valente, C., Turacchio, G., Bonsra, A.N., Du, G., et al. (2008). A role for phosphatidic acid in COPI vesicle fission yields insights into Golgi maintenance. *Nat. Cell Biol.* *10*, 1146–1153.

Reviewers' Comments:

Reviewer #1:

Remarks to the Author:

Overall, the authors have done an impressive job in revising the MS. Almost all my concerns have been satisfactorily addressed. The only concern that remains unaddressed is the request to perform RNA-seq in response to my comment 1. The authors justify this by the severe restrictions imposed by COVID-19, and I am willing to accept this explanation.

Therefore, I recommend acceptance after the minor comments below are properly addressed.

BTW: the rebuttal refers to Fig. 1e as a Western showing HIF1a stabilization. This seems to be a mistake in figure number.

1. The paper by Amelio et al (reference 65 in this MS) is mentioned only once, in a rather vague context of mutant p53 and ECM. However, the Amelio paper mainly focuses on the transcriptional effects of the interaction between mutant p53 and HIF1a, a theme that is closely related to the present MS. For the sake of fairness, this should be mentioned in the Discussion.

2. Line 324: please remove "and"

3. Line 441: Figure number (7) missing.

Reviewer #2:

Remarks to the Author:

The authors adequately addressed some of the major comments and made appropriate changes in the manuscript by including additional supplemental data as well as updating panels to the main figures. Good effort was made to perform some of the suggested experiments. Satisfactory new experiments were performed to demonstrate increased HIF1 interaction with mut-p53 under hypoxic conditions, and recruitment of mut-p53 to the miR-30d promoter. Notably, the authors were able to show that in tissue culture cells and in both human and mouse tissues, the protein levels of PDIA5, Sec24A, and GM130 were affected by miR-30d expression. Convincing new evidence in human breast tissue expressing mut-p53 stained for Sec24A and β -COP sufficiently supports the authors' findings that the mut-p53/miR-30d axis affects the secretory machinery. Discussion of the data and interpretation of the results has improved.

Experiments which were not performed due to COVID-19 lockdown, such as - the EndoH experiment for E-cad and other experiments to interrogate trafficking rate to plasma membrane, or identification of specific cargoes for which secretion rates are reduced - would have been insightful but less critical to the author's overall conclusions in the paper. This work opens up many avenues to pursue in understanding of the mechanistic pathway in which mut-p53 and HIF1 oncogenes regulate Golgi structure and function.

Overall, the data describes a novel p53/HIF1a/miR-30d axis that impacts synthesis, trafficking, and secretion of numerous proteins essential for tumor growth and metastatic colonization.

There still remains a few specific points that should be addressed:

1. Although the authors speculate that miR-30d stabilization of microtubules as mechanism for enhanced Golgi vesiculation and accelerated protein trafficking, the data to support this is limited and perhaps probing for difference in Ac-Tubulin in the conditions presented in Supplementary Fig 6l would strengthen the immunofluorescence observations from Fig 3a.

2. If local activation of PKD is central to the mechanistic hypothesis for the role of DGKZ in affecting Golgi structure, then Fig 4g is not sufficient. An additional method of detecting activated PKD1 at the Golgi would strengthen the author's claim. A simple IB for Golgi localized phospho-PKD1 for the same conditions tested in the IF is recommended.

3. The quantitation for the RUSH assay (Fig 3d) which measures trafficking from ER to Golgi is appreciated. Statistical significance demarcation in the graph is missing.

4. New Fig 1-h has a bar for statistical analysis from control normoxia to control hypoxia but is

missing asterisk(s) to denote significance.

5. Fig. 5b has some overlapping words on the label.

Reviewer #3:

Remarks to the Author:

The revised version of the manuscript has been significantly improved and is compelling with the concerns raised within my previous evaluation. Based on this, the manuscript is suitable for publication.

Reviewer #1 (Remarks to the Author):

Overall, the authors have done an impressive job in revising the MS. Almost all my concerns have been satisfactorily addressed. The only concern that remains unaddressed is the request to perform RNA-seq in response to my comment 1. The authors justify this by the severe restrictions imposed by COVID-19, and I am willing to accept this explanation. Therefore, I recommend acceptance after the minor comments below are properly addressed. BTW: the rebuttal refers to Fig. 1e as a Western showing HIF1a stabilization. This seems to be a mistake in figure number.

1. The paper by Amelio et al (reference 65 in this MS) is mentioned only once, in a rather vague context of mutant p53 and ECM. However, the Amelio paper mainly focuses on the transcriptional effects of the interaction between mutant p53 and HIF1a, a theme that is closely related to the present MS. For the sake of fairness, this should be mentioned in the Discussion.

We have modified citation of this reference in the Discussion, mentioning focus on the transcriptional partnership between mutant p53 and HIF1a (pag.14)

2. Line 324: please remove “and”. This has been removed.
3. Line 441: Figure number (7) missing. This has been added.

Reviewer #2 (Remarks to the Author):

The authors adequately addressed some of the major comments and made appropriate changes in the manuscript by including additional supplemental data as well as updating panels to the main figures. Good effort was made to perform some of the suggested experiments. Satisfactory new experiments were performed to demonstrate increased HIF1 interaction with mut-p53 under hypoxic conditions, and recruitment of mut-p53 to the miR-30d promoter. Notably, the authors were able to show that in tissue culture cells and in both human and mouse tissues, the protein levels of PDIA5, Sec24A, and GM130 were affected by miR-30d expression. Convincing new evidence in human breast tissue expressing mut-p53 stained for Sec24A and β -COP sufficiently supports the authors' findings that the mut-p53/miR-30d axis affects the secretory machinery. Discussion of the data and interpretation of the results has improved.

Experiments which were not performed due to COVID-19 lockdown, such as - the EndoH experiment for E-cad and other experiments to interrogate trafficking rate to plasma membrane, or identification of specific cargoes for which secretion rates are reduced – would have been insightful but less critical to the author's overall conclusions in the paper. This work opens up many avenues to pursue in understanding of the mechanistic pathway in which mut-p53 and HIF1 oncogenes regulate Golgi structure and function.

Overall, the data describes a novel p53/HIF1a/miR-30d axis that impacts synthesis, trafficking, and secretion of numerous proteins essential for tumor growth and metastatic colonization.

There still remains a few specific points that should be addressed:

1. Although the authors speculate that miR-30d stabilization of microtubules as mechanism

for enhanced Golgi vesiculation and accelerated protein trafficking, the data to support this is limited and perhaps probing for difference in Ac-Tubulin in the conditions presented in Supplementary Fig 6l would strengthen the immunofluorescence observations from Fig 3a.

Although the role of microtubule stabilization in the effect that the mut-p53/miR-30d axis has on secretion was not a major focus of our work, in the previous version of the MS we have addressed and clarified this point with several approaches. Our data clearly show that microtubule (MT) stabilizing agents increase Golgi vesiculation and protein secretion in conditions where secretion is impaired due to inhibition of miR-30d. We observed that miR-30d induces microtubule stabilization, and to add further evidence in support to this finding, as suggested by the reviewer we have now provided a new supplementary Fig. 6l, demonstrating that ectopic expression of miR-30d increases the level of Acetylated-Tubulin.

2. If local activation of PKD is central to the mechanistic hypothesis for the role of DGKZ in affecting Golgi structure, then Fig 4g is not sufficient. An additional method of detecting activated PKD1 at the Golgi would strengthen the author's claim. A simple IB for Golgi localized phospho-PKD1 for the same conditions tested in the IF is recommended.

In our work we show that, by targeting DGKZ, miR-30d leads to DAG (diacyl-glycerol) accumulation, which impacts on Golgi structure and function in different ways. To demonstrate the impact of DAG accumulation on downstream signaling, we monitored local activation of PKD due to DAG, by confocal imaging analysis of activated PKD localized at Golgi apparatus.

As further evidence in support to this finding, we have now added an analysis of activated PKD (phospho-PKD) in cells silenced for mut-p53 and observed a reduction of phospho-PKD while either miR-30d overexpression or DGKZ knock-down reversed this effect (new Suppl. Figure 6h). These data are consistent with the results of the previous confocal imaging approach.

3. The quantitation for the RUSH assay (Fig 3d) which measures trafficking from ER to Golgi is appreciated. Statistical significance demarcation in the graph is missing. We have added RUSH statistical methods and demarcation of statistical significance to the figure.

4. New Fig 1-h has a bar for statistical analysis from control normoxia to control hypoxia but is missing asterisk(s) to denote significance.

The asterisks have been added to Fig 1h.

5. Fig. 5b has some overlapping words on the label.

This has been adjusted.

Reviewer #3 (Remarks to the Author):

The revised version of the manuscript has been significantly improved and is compelling with the concerns raised within my previous evaluation. Based on this, the manuscript is suitable for publication.